# In vivo neutralization of coral snake venoms with an oligoclonal nanobody mixture in a murine challenge model

Melisa Benard-Valle [1], Yessica Wouters[1], Anne Ljungars [1], Giang Thi Tuyet Nguyen [1], Shirin Ahmadi[1], Tasja Wainani Ebersole[1], Camilla Holst Dahl[1], Alid Guadarrama-Martínez [2], Frederikke Jeppesen[1], Helena Eriksen [1], Gibran Rodríguez-Barrera [2], Kim Boddum [3], Timothy Patrick Jenkins [1], Sara Petersen Bjørn[1], Sanne Schoffelen[1], Bjørn Gunnar Voldborg[1], Alejandro Alagón[2] & Andreas Hougaard Laustsen [1] ✉

Oligoclonal mixtures of broadly-neutralizing antibodies can neutralize complex compositions of similar and dissimilar antigens, making them versatile tools for the treatment of e.g., infectious diseases and animal envenomations. However, these biotherapeutics are complicated to develop due to their complex nature. In this work, we describe the application of various strategies for the discovery of cross-neutralizing nanobodies against key toxins in coral snake venoms using phage display technology. We prepare two oligoclonal mixtures of nanobodies and demonstrate their ability to neutralize the lethality induced by two North American coral snake venoms in mice, while individual nanobodies fail to do so. We thus show that an oligoclonal mixture of nanobodies can neutralize the lethality of venoms where the clinical syndrome is caused by more than one toxin family in a murine challenge model. The approaches described may find utility for the development of advanced biotherapeutics against snakebite envenomation and other pathologies where multi-epitope targeting is beneficial.

By definition, broadly-neutralizing antibodies can neutralize several similar antigens that are not identical but cause the same physiological effects and may therefore be useful for the treatment of many infectious diseases[1-3], but certainly also snakebite envenomations, where multiple similar (and dissimilar) toxins in a venom need to be neutralized. In the Americas, over 2400 people fall victim to coral snake envenomations annually[4]. This group of snakes is highly diverse, and different species possess venoms that significantly differ in their overall composition[5]. However, the key targets to neutralize for therapeutic antibodies are the toxins relevant for the envenomation of mammals, including humans. Across coral snake venoms, these are relatively few and belong to only two protein subfamilies, namely

neurotoxic phospholipases $A_2$ (PLA$_2$s) and $\alpha$-neurotoxins ($\alpha$NTxs) from the three-finger toxin family (3FTxs)[5-7]. When these toxins are injected into mammalian prey and victims, they exert neurotoxic effects that manifest clinically as flaccid paralysis of skeletal muscles, which, if left untreated, can be fatal as this condition may progress to respiratory failure[8,9]. Currently, the only available specific treatments for envenomed patients are antivenoms that consist of polyclonal antibodies isolated from the plasma of hyperimmunized animals[10]. While these antivenoms have saved countless lives, they unfortunately suffer from several drawbacks, including a limited capacity to cross-neutralize venoms from different coral snake species[9,11], batch-to-batch variation, and a low content of therapeutically active

[1]Department of Biotechnology and Biomedicine, Technical University of Denmark, DK-2800 Kongens, Lyngby, Denmark. [2]Departamento de Medicina Molecular y Bioprocesos, Instituto de Biotecnología, Universidad Nacional Autónoma de México, Avenida Universidad 2001, Cuernavaca, Mor 62210, México. [3]Sophion Bioscience, DK-2750 Ballerup, Denmark. ✉e-mail: ahola@bio.dtu.dk

antibodies[12,13]. In comparison with many other antivenoms, such as those for viper envenomations, the issue with the low amount of neutralizing antibodies is particularly relevant for coral snake antivenoms, as the low abundance and limited immunogenicity of some of the medically most relevant toxins in coral snake venoms (i.e., αNTxs)[14,15] make it difficult to raise neutralizing antibodies via the animal immunization process used for traditional antivenom manufacturing[15,16]. Therefore, very high doses of antivenom are typically needed to treat severe envenomations, which further increases the risk of adverse reactions due to the heterologous nature of the antivenom antibodies.

To address the abovementioned issues, several researchers aim towards developing new types of antivenom products, e.g., recombinant antivenoms. One approach that has proven promising is the generation of recombinant antibodies against key venom toxins using phage display technology[17–19]. This methodology enables the discovery of specific antibodies against the most medically relevant toxins, regardless of their abundance within the venoms or immunogenicity[20]. Phage display technology can furthermore facilitate the discovery of cross-neutralizing antibodies (antibodies that can neutralize more than one toxin isoform) through cross-panning strategies[17,19,21] or selections against recombinantly produced consensus antigens (i.e., antigens designed to represent the 'average' of several different proteins)[22]. Finally, it has been shown that monoclonal antibodies can be combined as carefully generated oligoclonal mixtures, allowing for the neutralization of multiple toxins by a single cocktail[18]. These discoveries have further led to the speculation that recombinant antivenoms with very broad neutralization capacity (i.e., polyvalent recombinant antivenoms) can be developed by preparing oligoclonal mixtures of individual cross-neutralizing monoclonal antibodies[23] and that this may be a promising approach to develop a new type of affordable envenomation therapies[24,25].

Replacing polyclonal antibodies purified from the plasma of immunized animals with recombinant oligoclonal antibody mixtures may have the potential to significantly reduce batch-to-batch variation, to neutralize all medically relevant toxins in the targeted venoms, and to ensure a high therapeutic antibody content in the recombinant products. To date, work on recombinant monoclonal antibodies has primarily focused on using human monoclonal immunoglobulin G (IgG) antibodies, which are highly specific and have long half-lives in circulation, but have limited stability ex vivo and are relatively expensive to manufacture compared to other types of recombinant binding proteins[25]. As an alternative, other therapeutically promising antibody scaffolds, such as camelid single-domain antibodies ($V_H$Hs, also known as nanobodies) have come into focus[26]. $V_H$Hs are derived from heavy-chain-only antibodies present in Camelidae and are characterized by possessing similarly high affinities and specificities as IgG antibodies. They are more stable at high temperatures and extreme pH[27] than IgGs, and they can generally be expressed in a large scale at a lower cost[25]. Their small size (12–15 kDa) increases their ability to penetrate deep tissues but has the drawback of resulting in a short serum half-life. However, if needed, the circulation half-life can be optimized through protein engineering techniques, such as fusion with a human Fc domain[26,28] or assembly into larger protein architectures[19,29] (although seldom without affecting other parameters such as tissue penetration).

To investigate the utility of $V_H$Hs against snakebite envenomation, in this study, we aimed to discover cross-neutralizing $V_H$Hs against the key toxins in coral snake venoms to enable the preparation of an oligoclonal $V_H$H mixture that could be used to treat coral snake envenomation. To this end, we used an immune $V_H$H phage display library from one alpaca and one llama immunized with multiple elapid snake venoms for discovery of $V_H$Hs targeting the medically most important toxins in coral snake venoms. To further facilitate the discovery of cross-neutralizing $V_H$Hs, we utilized a recombinant consensus antigen

representing αNTxs and a native representative neurotoxic PLA$_2$ as antigens. Using both a rodent model involving pre-incubation of venom and $V_H$Hs, as well as a rodent model mimicking a real-life snakebite envenomation, where venom is first injected subcutaneously (s.c.), followed by administration of $V_H$Hs intravenously (i.v.) (a.k.a. a rescue model), we demonstrated that the discovered $V_H$Hs can neutralize the lethality of coral snake neurotoxins. Furthermore, we showed that an oligoclonal mixture of only two cross-neutralizing $V_H$Hs, even in a monovalent format, can neutralize the lethality of the whole venoms of *Micrurus fulvius* (Eastern coral snake, US) and *Micrurus diastema* (variable coral snake, Mexico) performing comparably to the existing plasma derived antivenom (Coralmyn), while individual $V_H$Hs fail to do so on their own. Finally, we explore the utility of two alternative antibody constructs (a homodimeric $V_H$H-$V_H$H construct and a $V_H$H-Fc construct) and conclude that these do not perform better than the simpler monovalent $V_H$Hs.

## Results
### Camelid immunization and $V_H$H phage display library generation
To facilitate the discovery of cross-neutralizing $V_H$Hs, one alpaca and one llama were immunized with increasing doses of a mixture of 18 elapid venoms over 16-weeks (Supplementary Table 1), followed by construction of $V_H$H displaying phage libraries. Blood samples were collected from the animals before immunization (day 0) and preceding venom inoculation on each injection day. Subsequently, an Enzyme-Linked Immunosorbent Assay (ELISA) was employed to analyze the antibody responses over time against each of the individual venoms used for immunization. Antibody binding signals increased over time for all venoms, and both camelids showed comparable immune responses against the same venoms. For example, lower binding signals were observed against *Dendroaspis* venoms compared to *Naja* venoms (Fig. 1). After 8 and 16 weeks of immunization, one $V_H$H-displaying phage library was generated from each animal, all larger than 3·10$^8$ individual clones with insert rates > 85%. As serum samples from the two animals showed a comparable immune response, the phage libraries were mixed, resulting in one library from 8 weeks and one from 16 weeks of immunization.

### Phage display selection campaigns and screening of $V_H$H binders
To enable the preparation of an oligoclonal cocktail for the treatment of coral snake envenomation, we focused our discovery efforts of $V_H$Hs against the medically most relevant toxins in coral snake venoms namely neurotoxic PLA$_2$s, from the PLA$_2$ family, and αNTxs from the 3FTx family. The decision to use this immune phage display library to discover $V_H$Hs against coral snake venom toxins was based on the high sequence similarity observed between toxins in coral snake venoms and in venoms from other elapid snake species such as *Naja* and *Dendroaspis*. An alignment of short chain αNTxs from these genera can be found in Supplementary Fig. 1. The 16-week library was panned against purified toxins selected to facilitate the discovery of cross-neutralizing $V_H$Hs. These included two wildtype PLA$_2$s purified from the venom of *M. fulvius* (denominated PLA2N and PLA2O[6]), two recombinantly expressed native αNTxs from the venoms of *Micruroides euryxanthus* and *M. diastema* (named rEury[30] and rDH[31], respectively), and a recombinantly expressed short-chain consensus αNTx (named scNTx[32]). After two or three consecutive rounds of selection, five enriched libraries (TPL0637, TPL0638, TPL0622, TPL0623, and TPL0629) were chosen for further studies based on the enrichment of the phage pools (i.e., the number of colony-forming units in the performed selection outputs compared to a negative selection run in parallel without any antigen) (Supplementary Figs. 3A and 4A). The selected libraries were subcloned into a modified pHEN6 expression vector for soluble $V_H$Hs and transformed into BL21 (DE3) cells[33].

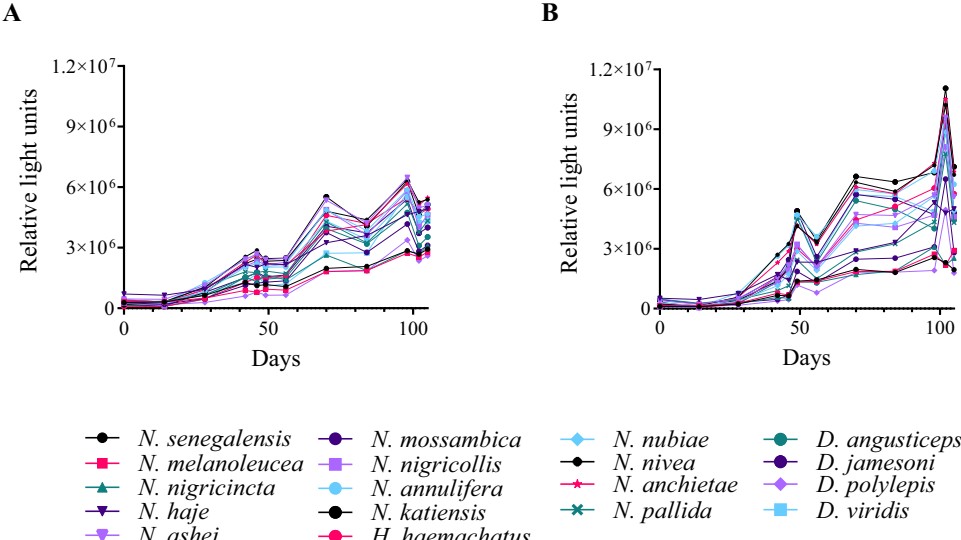

Legend:
- N. senegalensis ● N. mossambica ◆ N. nubiae ● D. angusticeps
- N. melanoleuca ■ N. nigricollis ● N. nivea ● D. jamesoni
- N. nigricincta ▲ N. annulifera ★ N. anchietae ◆ D. polylepis
- N. haje ▼ N. katiensis ✕ N. pallida ■ D. viridis
- N. ashei ● H. haemachatus

**Fig. 1 | Antibody responses of two immunized camelids over time.** Antibody binding signals observed for serum samples collected at different time points from two camelids immunized with a mixture of 18 elapid venoms. **A** Response of llama 0406 to the 18 venoms included in the immunization mixture. **B** Response of alpaca 0541 to the 18 venoms included in the immunization mixture. Values correspond to the means of two replicates ($n = 2$). The signal at day 0 represents the response of the preimmune sera. Source data are provided as a Source Data file.

Monoclonal $V_HHs$ from individually picked clones were expressed and binding to their cognate target was assessed using an expression-normalized capture Dissociation-Enhanced Lanthanide Immunoassay (DELFIA) (Supplementary Figs. 2B, C and 3B–D). In agreement with the enrichment of the phage pools, a higher fraction of binding clones was observed from the selection campaign performed using the consensus scNTx (Supplementary Fig. 3D) compared to the campaigns using native toxins as antigens (Supplementary Fig. 3B, C). This could indicate that the epitope(s) of the consensus toxin have more in common with the epitope(s) of the toxins used for immunization compared to the native coral snake toxins used in the experiment (rEury and rDH). Therefore, in this experiment, the use of this consensus toxin was advantageous over the use of the native toxins. Based on the screening results, 24 and 49 clones with high binding signals to $PLA_2$ and αNTxs, respectively, were selected for sequencing. Out of the 79 sequenced clones, 19 anti-$PLA_2$ and 17 anti-αNTx $V_HHs$ showed unique sequences, which could be further clustered in 4 and 7 families, respectively, based on sequence similarity of the complementarity-determining regions (CDRs).

The three unique $V_HH$ clones targeting either $PLA_2s$ or αNTxs (Supplementary Fig. 5) that showed the highest binding signals in the expression-normalized capture DELFIA were further analyzed in dose-response experiments to assess cross-reactivity to $PLA_2s$ and αNTxs from other snake genera. The three anti-$PLA_2$ $V_HHs$ all showed comparable binding to $PLA_2$- and $PLA_2$-containing fractions from the venoms of *Naja melanoleuca* (Nm15), *Naja nigricollis* (Nn19), and *Hemachatus haemachatus* (Hh3) with TPL0638_01_C09 showing the highest binding signal. No binding to a structurally different $PLA_2$ from *Echis pyramidum* was observed (Supplementary Fig. 5). Similarly, the three αNTx-targeting $V_HHs$ also bound to αNTxs from the venoms of *M. diastema* (DH), *Naja haje* (Nh1), *Dendroaspis viridis* (Dv1), and *H. haemachatus* (Hh1) with TPL0629_01_D11 showing the highest binding signal and the lowest $EC_{50}$-value (Supplementary Fig. 6). No binding to α-cobratoxin, a long chain αNTx from the 3FTx family purified from the venom of *Naja kaouthia*, was observed. Taken together, the high affinity of the discovered $V_HHs$ to $PLA_2s$ and αNTxs from different snake genera demonstrates that they possess cross-reactive binding across their respective toxin (sub)families.

## Analysis of $V_HH$ binding kinetics with biolayer interferometry

To characterize the binding kinetics of the selected $V_HH$ clones against each toxin group, biolayer interferometry (BLI) experiments were performed using $PLA_2N$ and αNTx DH as antigens. The data were fitted to a 1:1 binding model, which assumes a single $V_HH$ molecule binds to a single toxin molecule. All the tested $V_HHs$ showed high affinity, with $K_D$ values in the pM range for the three anti-$PLA_2$ $V_HHs$ and in the nM range for the three anti-αNTx $V_HHs$. Four of the six monovalent $V_HHs$ showed very slow dissociation rates ($k_{off} < 5.5 \cdot 10^{-4}$ s$^{-1}$), which indicates that they remain bound to their respective toxin for a long time period (Fig. 2, Table 1).

## Neutralization of $PLA_2$ enzymatic activity

To assess the potential cross-neutralizing capacity of the $PLA_2$-binding $V_HHs$, an in vitro enzymatic $PLA_2$ activity neutralization assay was performed using a commercial kit and three of the $PLA_2s$ and $PLA_2$-containing fractions that the $V_HHs$ showed binding to ($PLA_2N$, Nn19, and Hh3). The $V_HHs$ were evaluated at a 1:20 toxin to $V_HH$ molar ratio (0.07 μM and 1.4 μM). All three anti-$PLA_2$ $V_HHs$ reduced the enzymatic activity of $PLA_2N$ as well as the $PLA_2$ from the venom of *N. nigricollis* (Nn19) but showed no inhibition of the activity of Hh3 (*H. haemachatus*) (Fig. 3A). The $V_HHs$ alone did not show any $PLA_2$ activity. This result demonstrates the neutralization capacity of the $V_HHs$ for $PLA_2s$ beyond those in the venoms of *Micrurus*. Further work is necessary to assess the utility of these $V_HHs$ to neutralize the toxic effects of *H. haemachatus* $PLA_2s$, as well as $PLA_2s$ from other snake genera.

## Neutralization of αNTx-mediated blocking of the nicotinic acetylcholine receptor (nAChR)

The ability of the discovered anti-αNTx $V_HHs$ to neutralize αNTx-mediated blocking of the muscle type nAChR function was evaluated using whole-cell patch clamp recordings of rhabdomyosarcoma cells endogenously expressing the nAChR, which were exposed to purified αNTx DH or the consensus toxin scNTx at a concentration resulting in approximately 80% inhibition ($IC_{80}$) (Fig. 3B). When tested against the αNTx DH, two of the $V_HHs$ could completely neutralize the αNTx activity at $V_HH$ concentrations higher than 45 nM, corresponding to molar ratios above 1:3 between the toxin and $V_HH$. Approximately half

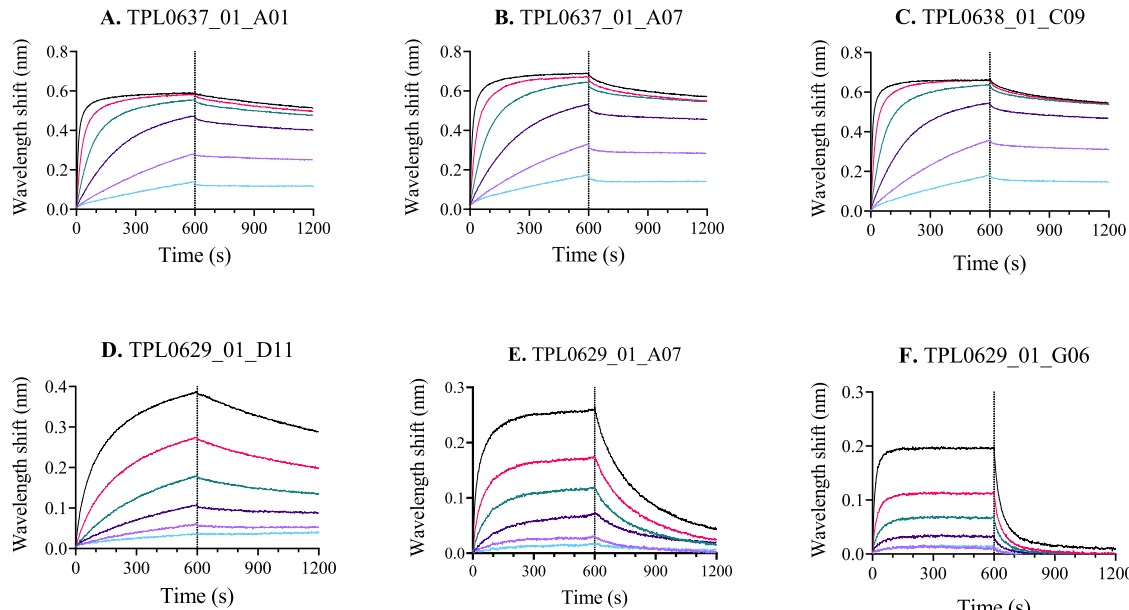

**Fig. 2 | Binding kinetics of $V_H$Hs to purified toxins.** Biotinylated toxins captured on streptavidin biosensors were dipped in decreasing concentrations of each of the $V_H$Hs, followed by dissociation in kinetics buffer. Binding data were fitted using a 1:1 model. The colors represent the different $V_H$H concentrations: black is 200 nM, pink is 67 nM, green is 22 nM, dark purple is 7.4 nM, light purple is 2.5 nM, and cyan is 0.8 nM. **A**–**C** Anti-PLA₂ $V_H$Hs binding PLA₂N. **D**–**F** Anti-αNTx $V_H$Hs binding αNTx DH. Source data are provided as a Source Data file.

**Table 1 | Biolayer interferometry binding parameters of $V_H$Hs to purified toxins**

|   | Toxin | $V_H$H | $K_D$ (M) | $K_D$ error | $k_{on}$ (1/Ms) | $k_{on}$ error | $k_{off}$ (1/s) | $k_{off}$ error | $R^2$ |
|---|---|---|---|---|---|---|---|---|---|
| **A** | PLA₂N | TPL0637_01_A01 | $4.76 \cdot 10^{-10}$ | $4.51 \cdot 10^{-12}$ | $5.27 \cdot 10^{+05}$ | $1.98 \cdot 10^{+03}$ | $2.51 \cdot 10^{-04}$ | $2.18 \cdot 10^{-06}$ | 0.997 |
| **B** | PLA₂N | TPL0637_01_A07 | $5.25 \cdot 10^{-10}$ | $7.26 \cdot 10^{-12}$ | $5.27 \cdot 10^{+05}$ | $3.13 \cdot 10^{+03}$ | $2.77 \cdot 10^{-04}$ | $3.46 \cdot 10^{-06}$ | 0.992 |
| **C** | PLA₂N | TPL0638_01_C09 | $4.10 \cdot 10^{-10}$ | $3.74 \cdot 10^{-12}$ | $6.73 \cdot 10^{+05}$ | $3.13 \cdot 10^{+03}$ | $2.76 \cdot 10^{-04}$ | $2.27 \cdot 10^{-06}$ | 0.996 |
| **D** | αNTx DH | TPL0629_01_D11 | $6.94 \cdot 10^{-09}$ | $5.92 \cdot 10^{-11}$ | $7.47 \cdot 10^{+04}$ | $4.41 \cdot 10^{+02}$ | $5.18 \cdot 10^{-04}$ | $3.19 \cdot 10^{-06}$ | 0.995 |
| **E** | αNTx DH | TPL0629_01_A07 | $1.46 \cdot 10^{-08}$ | $1.36 \cdot 10^{-10}$ | $2.51 \cdot 10^{+05}$ | $2.23 \cdot 10^{+03}$ | $3.66 \cdot 10^{-03}$ | $1.05 \cdot 10^{-05}$ | 0.992 |
| **F** | αNTx DH | TPL0629_01_G06 | $6.91 \cdot 10^{-08}$ | $1.11 \cdot 10^{-09}$ | $2.63 \cdot 10^{+05}$ | $4.00 \cdot 10^{+03}$ | $1.82 \cdot 10^{-02}$ | $9.12 \cdot 10^{-05}$ | 0.994 |

Binding data were fitted using a 1:1 model. Letters A–F correspond to kinetic graphs in Fig. 2.

of the toxins' activity was neutralized at a 1:1 toxin to $V_H$H molar ratio (corresponding to a $V_H$H concentration of 15 nM). The three $V_H$Hs performed better against the consensus toxin scNTx where full neutralization was observed at a 1:1 toxin to $V_H$H molar ratio (5 nM of $V_H$H) for two of the $V_H$Hs.

## In vivo neutralization of purified toxins

Based on the in vitro binding and neutralization data, the three most promising $V_H$Hs against PLA₂s and αNTxs were assessed for neutralization of lethality with purified toxins via intravenous (i.v.) injection of the preincubated toxin and $V_H$H (preincubation experiments) and i.v. injection of $V_H$H after envenomation using subcutaneous (s.c.) injection (rescue experiments). The $V_H$Hs were evaluated for their ability to neutralize PLA₂N and wild-type αNTx DH induced lethality, respectively. As negative controls, two groups of mice were injected with 3 median lethal doses (LD₅₀s) of PLA₂N or αNTx DH preincubated with an isotype $V_H$H. As expected, no neutralization or delay of time of death was observed in these groups. Also, a group of mice was injected i.v. with the highest dose evaluated of each $V_H$H. None of these mice showed any signs of adverse reactions (in vivo data are shown in Supplementary Tables 2 and 3).

Neutralization of PLA₂-induced lethality was evaluated using 3 LD₅₀s of PLA₂N, purified from the venom of *M. fulvius*. The LD₅₀ of this toxin was determined to be 10.3 μg/mouse when given i.v. and 34.6 μg/

mouse with s.c. administration. The three anti-PLA₂ $V_H$Hs completely neutralized PLA₂N-induced lethality when preincubated with PLA₂N and injected i.v. at a 1:1 toxin to $V_H$H molar ratio (Fig. 4A). Conversely, when the $V_H$Hs were injected i.v. immediately after s.c. injections of PLA₂N (1:2.5 toxin to $V_H$H molar ratio), only TPL0637_01_A07 neutralized PLA₂N-induced lethality in all three mice, while the other two rescued 1 of the 3 mice (Fig. 4B).

Even though αNTxs are in relatively low abundance in Mexican coral snake venoms, it has been shown, using polyclonal sera from immunized horses, that the presence of neutralizing antibodies against these toxins is necessary for the neutralization of several of these venoms, including *M. diastema*[14]. The LD₅₀ of wild-type αNTx DH from the venom of *M. diastema* was determined to be 2.0 μg/mouse when given i.v. and 4.8 μg/mouse with s.c. administration and neutralization of lethality by $V_H$Hs was evaluated using 3 LD₅₀s of the toxin. TPL0629_01_D11 and TPL0629_01_A07 prevented lethality in all the mice when preincubated with toxin at a 1:2.5 molar ratio between toxin and $V_H$H followed by i.v. injection of the mixture (Fig. 4C). In the same type of experiment, TPL0629_01_G06 did not prevent death of mice using the same molar ratio between toxin and $V_H$H. Although survival of the mice was identical for the first two $V_H$Hs, some signs of paralysis of the mice were observed when using TPL0629_01_A07 for neutralization and therefore the following experiments were only performed with TPL0629_01_D11. When 3 LD₅₀s of the toxin were injected

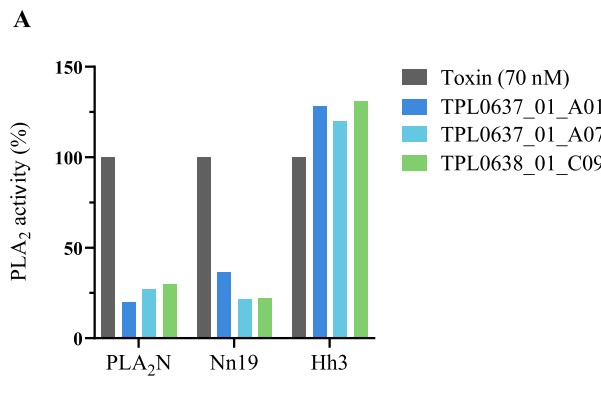

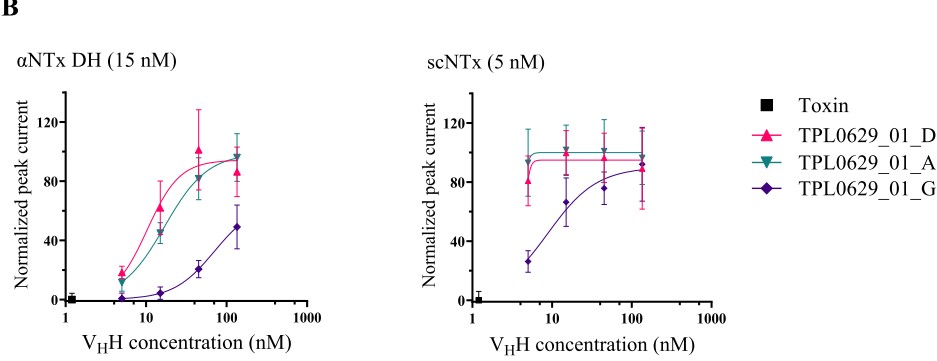

**Fig. 3 | In vitro neutralization of toxin activity by V_HHs. A** Inhibition of PLA_2 enzymatic activity by V_HHs. The maximal enzyme activity observed with toxin alone was set to 100%. The normalized enzymatic activity of PLA_2s from various elapid genera (PLA_2N, Nn19, and Hh3) preincubated for 30 min at RT with anti-PLA_2 V_HHs at a 1:20 toxin to V_HH molar ratio. Bars represent the mean of two replicates.

**B** Neutralization of αNTx-mediated blocking of muscle-type nAChR current. Dose-response curves from patch clamp experiments with increasing concentrations of V_HHs to prevent the blocking of nAChR by 15 nM αNTx DH or 5 nM scNTx. Error bars represent standard deviation of independent cells performed in a 384-well plate. $n = 16$. Source data are provided as a Source Data file.

s.c. followed by immediate i.v. injection of the V_HH, all mice were rescued using a 1:10 toxin to V_HH molar ratio (Fig. 4D). To evaluate if a bivalent version of TPL0629_01_D11 would allow for a lower molar excess of the V_HH to be used for neutralization, two new constructs, a bivalent V_HH construct, where two V_HHs are connected by a GS-linker, and a V_HH-Fc, in which the V_HH is fused to a human Fc domain, were produced and evaluated in vivo. To confirm that both binding sites of the bivalent constructs are available for antigen binding, the constructs were analyzed in BLI with immobilized toxin as described in Materials and Methods (Section 4.10). Both constructs showed an increased avidity for αNTx DH (0.53 nM for the bivalent V_HH and 3.69 nM for the V_HH-Fc) compared to the monovalent V_HH (6.94 nM) (Supplementary Fig. 7). For neutralization, a 1:1.25 molar ratio between toxin and V_HH-Fc or the bivalent V_HH construct (corresponding to the same ratio between toxin and binding sites for all evaluated constructs) was used. The V_HH and the V_HH-Fc showed similar results, delaying the time of death for about 8–10 h, while the bivalent V_HH construct only prolonged the survival to about 3 h (Fig. 5). As no benefit of using the bivalent constructs was observed, further work focused only on the monovalent V_HH construct.

**In vivo neutralization of whole coral snake venoms**
As previously mentioned, PLA_2s and αNTxs are determined to be the key targets in the venom of coral snakes. Based on the work of Vergara et al.[6], approximately 60% of the venom of *M. fulvius* is composed of PLA_2s and around 32% of 3FTxs (Supplementary Fig. 8). RP-HPLC analysis of the venom of *M. diastema* has shown a similar venom composition, with approximately 62% of the total protein content being PLA_2s and 22% 3FTxs (Supplementary Fig. 8). Here, it is relevant to note that toxins in these venoms are not limited to PLA_2N and αNTx

DH, but the venoms also contain other similar toxins that need to be neutralized to prevent complete venom-induced lethality. To prepare oligoclonal mixtures containing a minimal, but sufficient, number of V_HHs, the most potent of the discovered V_HHs, namely TPL0629_01_D11 (neutralizing αNTxs) and TPL0637_01_A07 (neutralizing PLA_2s) were included in two different mixtures. The molar ratios of the V_HHs in these mixtures were based on the molar ratio between PLA_2s and 3FTxs in the venoms (Supplementary Table 4). Thereafter, the oligoclonal mixtures were evaluated for their ability to neutralize the whole venoms from *M. fulvius* and *M. diastema*.

The LD_50s of the venoms were determined to be 6.0 μg/mouse for *M. fulvius* and 5.7 μg/mouse for *M. diastema*, using the i.v. route for administration. When 3 LD_50s of each venom were preincubated with each oligoclonal mixture using a 1:10 toxin to V_HH molar ratio prior to i.v. administration, the respective mixture was able to prevent lethality in all the mice injected with *M. fulvius* venom and two of the three mice injected with *M. diastema* venom. For comparison, a plasma-derived antivenom used for the treatment of coral snake envenomation in Mexico, Coralmyn, was included as a control at a similar dose as the oligoclonal mixtures (a 1:10 ratio between toxin and antibody binding sites). Coralmyn prevented lethality in all mice injected with *M. fulvius* venom, whereas the three mice injected with *M. diastema* venom died within 3 h. As a reference, the final doses of the oligoclonal mixtures used were 14.0 mg/kg for *M. fulvius* and 12.9 mg/kg for *M. diastema* compared to 45.5 mg/kg and 40.7 mg/kg for Coralmyn. Preincubation of both *M. fulvius* or *M. diastema* venom with either TPL0637_01_A07 (targeting PLA_2s) or TPL0629_01_D11 (targeting αNTxs) alone did not prevent lethality or prolong survival for any of the envenomed mice (Fig. 6). Overall, the data show that both of the oligoclonal mixtures possess a comparable neutralization capacity than the traditional

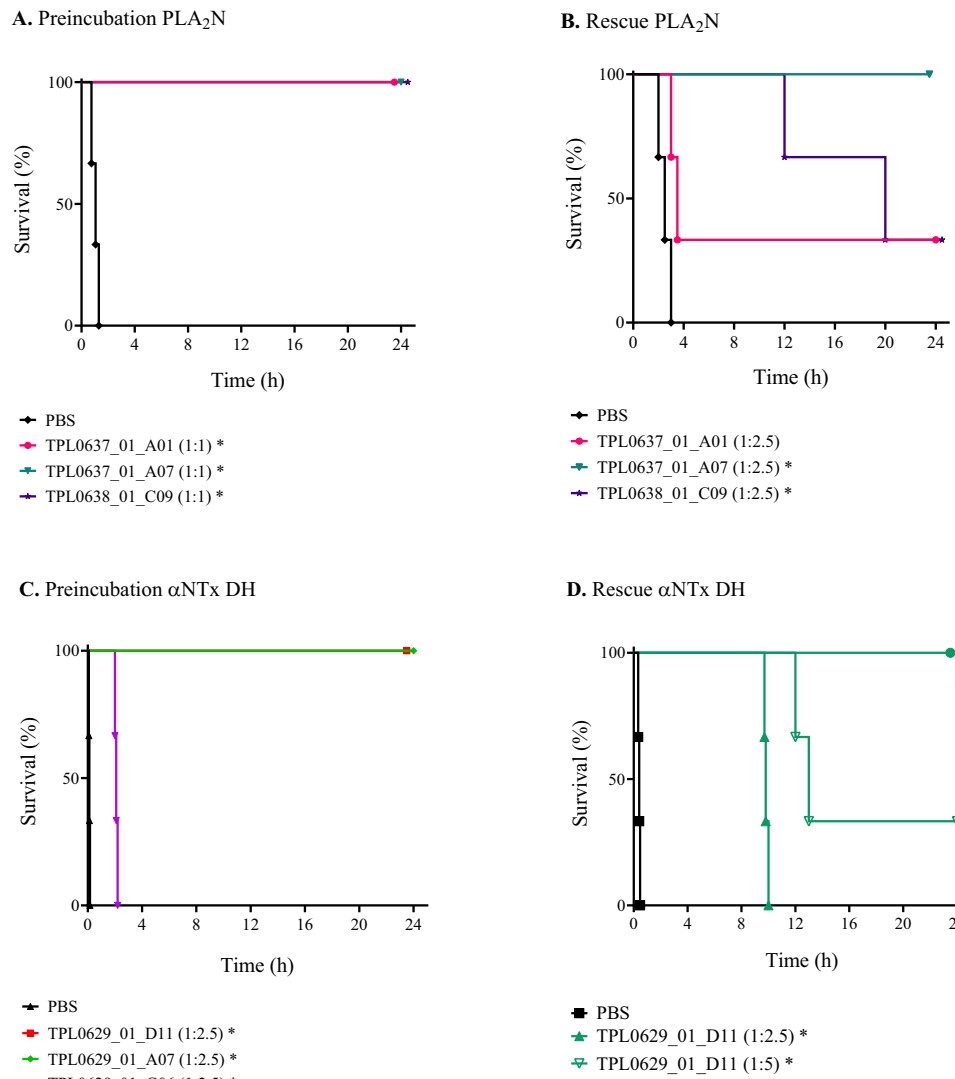

**Fig. 4 | Kaplan-Meier survival curves for mice challenged with PLA₂N or αNTx DH with or without adding V_HHs. A, C** 3 LD₅₀s of PLA₂N (**A**) or αNTx DH (**C**) were preincubated with either PBS or one of the V_HHs and injected in mice using the i.v. route. *n* = 3. **B, D** The mice were injected with 3 LD₅₀s of PLA₂N (**B**) or αNTx DH (**D**) using the s.c. route followed by immediate i.v. injection with PBS or one of the V_HHs. Toxin to V_HH molar ratio is presented in parentheses. *n* = 3. * Indicates a significant difference to PBS control (*P* < 0.05) in a Mantel-Cox log-rank test. Source data are provided as a Source Data file.

antivenom, Coralmyn, on the two tested venoms, indicating a possibly broader species coverage (in vivo raw data is shown in Supplementary Table 5). Further optimization and neutralization assays on other coral snake venoms could, however, likely be used to identify an even more optimal V_HH mixture for a final recombinant antivenom product.

## Discussion

The use of oligoclonal antibody mixtures to combat pathologies holds significant therapeutic promise, and such mixtures have already been tested against a number of indications, including several infectious diseases[34-36]. These advanced biologics derive their therapeutic potential from their ability to bind multiple targets or epitopes, which can be exploited to modulate intricate disease mechanisms or to neutralize several dissimilar pathogens. One of the indications for which oligoclonal antibody mixtures may be especially useful is snakebite envenomation. Here, complex mixtures of similar and dissimilar toxins need to be neutralized to prevent the onset of toxic effects[37,38]. To neutralize similar toxins from the same protein family, single cross-neutralizing monoclonal antibodies targeting variations of the same epitope can be applied[19]. However, developing therapies against

snakebite envenomation that cover multiple snake species, let alone a whole venom from a single snake species (which often contains several different toxin families of medical importance), large benefits emerge by utilizing oligoclonal antibody mixtures. For these to be effective, multiple cross-neutralizing monoclonal antibodies should be combined into broadly-neutralizing oligoclonal antibody mixtures that can neutralize several entire (sub)families of toxins[23]. In this study, we developed a prototype mixture based on cross-neutralizing V_HHs and demonstrated its ability to neutralize lethality induced by two North American coral snake venoms in a mouse model. Previous studies have shown that a single monoclonal antibody can neutralize the lethal effects of snake venom when lethality is caused by one predominant toxin[39,40], that such antibodies can be further developed to become broadly-neutralizing[19], and that oligoclonal antibodies can be used to neutralize multiple different toxins[18]. However, here, we go beyond the state of the art by demonstrating that the concepts from these previous studies can be merged and applied to a simpler antibody format, namely the V_HH format, thereby providing a proof of concept for a different type of antivenom[41]. This work also serves as a proof of principle for creating cross-neutralizing, multi-epitope targeting

oligoclonal antibody mixtures, which may find application beyond animal envenomations.

Specifically, in this study, we used existing data on the proteomic composition and toxicity of venoms from coral snakes[5–7,42–44] to focus our discovery efforts on the medically relevant presynaptic-acting PLA$_2$s and postsynaptic-acting αNTxs and establish a discovery approach for cross-neutralizing V$_H$Hs[17,21]. Using this approach, we discovered multiple V$_H$Hs and showed that three anti-αNTx V$_H$Hs bind a native αNTx from *M. diastema* with high affinity (Fig. 2, Table 1) as well as other αNTxs purified from the venoms of *N. haje*, *D. viridis*, and *H. haemachatus* (Supplementary Fig. 6), demonstrating their cross-reactive binding capabilities. Similarly, we showed that the three tested anti-PLA$_2$ V$_H$Hs can neutralize the enzymatic activity of PLA$_2$s from both *Micrurus* and *Naja* venoms in vitro, demonstrating their cross-

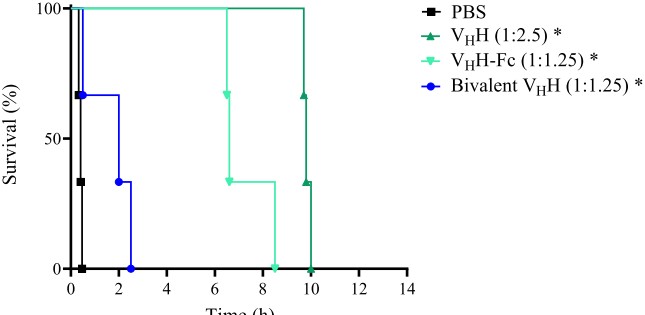

Rescue: V$_H$H constructs (αNTx DH)

**Fig. 5 | Kaplan-Meier survival curves for mice challenged with αNTx DH followed by injection of different V$_H$H constructs.** The mice were injected with 3 LD$_{50}$s of αNTx DH using the s.c. route followed by immediate i.v. injection with PBS, or the V$_H$H TPL0629_01_D11 as a monovalent V$_H$H construct, and as bivalent constructs (bivalent V$_H$H, or V$_H$H-Fc). Toxin to antibody construct molar ratios are presented in parentheses. Note that a molar ratio of 1:1.25 of the bivalent constructs is equivalent to a 1:2.5 molar ratio between toxin and binding sites. *n* = 3. * Indicates a significant difference to PBS control (*P* < 0.05) in a Mantel-Cox log-rank test. Source data are provided as a Source Data file.

neutralizing capacity (Fig. 3A). Not surprisingly, the discovered V$_H$Hs were unable to recognize a structurally different group of PLA$_2$s purified from the venom of the viper *E. pyramidum* (Supplementary Fig. 5). In rescue experiments using a rodent model, which mimics a real-life snakebite scenario[45], lethality induced by purified PLA$_2$N or αNTx DH could be neutralized by three of the identified cross-reactive V$_H$Hs individually (Fig. 4). An evaluation of one of the V$_H$Hs in three different formats, as monovalent V$_H$H, as a bivalent V$_H$H construct, and as a V$_H$H-Fc construct, for rescuing mice envenomed with αNTx revealed that, in this experimental setting, a larger bivalent antibody format did not exhibit any advantages in terms of prolonged survival compared to the simpler V$_H$H format. While the exact reason for this remains unknown, we speculate that the complex toxicokinetic and pharmacokinetic interplay between the toxins and the V$_H$H constructs plays a role. These findings, nevertheless, underscore the utility of V$_H$Hs for treating envenomation with venoms primarily composed of low molecular weight components, which have fast tissue absorption, such as PLA$_2$s and 3FTxs[46–48]. Finally, we generated two oligoclonal mixtures using the best neutralizing V$_H$H for each toxin family. We showed that a combination of only two V$_H$Hs could neutralize the lethality of complete coral snake venoms in vivo, but also that both V$_H$Hs were necessary to achieve this effect, as the two V$_H$Hs did not prevent lethality when administered individually (Fig. 5). Furthermore, we demonstrate that these two mixtures can cross-neutralize venoms from *M. fulvius* and *M. diastema*, unlike the antivenom that is currently used in the clinic (i.e., Coralmyn), which was only able to neutralize venom from *M. fulvius* (Fig. 6). In addition, the V$_H$H mixtures can be used at lower doses in terms of mg/kg compared to Coralmyn. It is relevant to note that this comparison does not consider the percentage of venom-specific antibodies present in polyclonal antivenoms, such as Coralmyn, since this is unknown. Given the abundance of PLA$_2$s and αNTxs similar to the ones used in this work in other North American coral snake venoms[5,30,49,50], our results support the feasibility of generating and applying recombinant oligoclonal antivenoms, composed of only a few V$_H$Hs, for neutralizing venoms from coral snake species in North America. Eventually, we predict that this approach could be employed in the design of genus-wide recombinant antivenoms or even polyvalent recombinant antivenoms covering several snake genera.

A. *Micrurus fulvius*

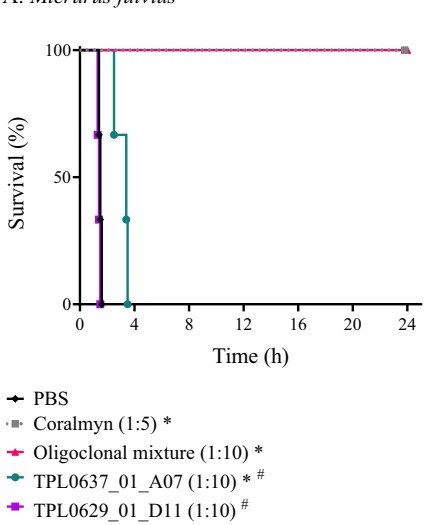

B. *Micrurus diastema*

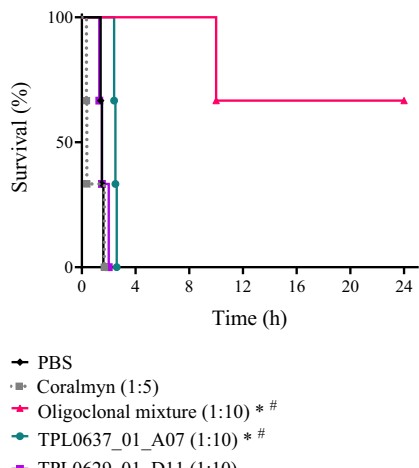

**Fig. 6 | Kaplan-Meier survival curves for mice challenged with whole venoms preincubated with an oligoclonal mixture of V$_H$Hs, individual V$_H$Hs, or the commercial antivenom, Coralmyn.** 3 LD$_{50}$s of venom from (**A**). *M. fulvius* or (**B**). *M. diastema*, were preincubated with either PBS, the individual V$_H$Hs, the relevant V$_H$H oligoclonal mixture prepared for the specific venom, or the polyclonal F(ab')$_2$-

based antivenom, Coralmyn. Approximate toxin to V$_H$H or F(ab')$_2$ molar ratios are shown in parentheses. Calculations are based on total protein content in the Coralmyn and oligoclonal mixtures. *n* = 3. * Indicates a significant difference to PBS control (*P* < 0.05). # Indicates a significant difference to Coralmyn (*P* < 0.05) in a Mantel-Cox log-rank test. Source data are provided as a Source Data file.

While the work presented here has promise, it still has some limitations. For example, this study did not attempt the neutralization of complete coral snake venoms in a rescue setting due to the high $V_H$H concentration required. To reduce the amount of $V_H$Hs needed for neutralization, future work should focus on the discovery of anti-αNTx $V_H$Hs with higher affinity to achieve neutralization at lower molar ratios similar to what has been seen previously with IgGs[19,39]. Our data showed that a higher affinity, specifically a slower dissociation rate between the $V_H$H and αNTxs, correlates with better neutralizing properties (Figs. 2 and 3, Table 1). In addition to the discovery of new $V_H$Hs with higher affinity, the current $V_H$Hs could potentially gain improved affinity through Bayesian optimization or random mutagenesis of the CDR regions[51,52]. Moreover, treatment of envenomation caused by more complex snake venoms, particularly those with high molecular weight enzymatic toxins, might require antibody formats with a longer circulation half-life, which can be achieved by leveraging different antibody scaffolds[12]. A final antivenom product could potentially combine various antibody formats and small molecule enzymatic inhibitors for optimal pharmacokinetic properties[12,53]. Finally, before any new type of antivenom product can be widely deployed, it will be paramount to establish how well the findings in this study and others translate to the clinical setting.

In the present study, we succeeded in discovering neutralizing $V_H$Hs against coral snake toxins from a $V_H$H library made from a llama and an alpaca that were not immunized with coral snake venoms but only venoms from African cobras and mambas. The data presented here thus demonstrates that it is possible to discover para-specific $V_H$Hs with such a library, and we predict that it could also be used to discover $V_H$Hs against toxins from several other snake genera within the elapid snake family. If this hypothesis is correct, it indicates that the development of therapeutically beneficial recombinant antivenoms may be more facile than previously expected.

Beyond snakebite envenomation, the discovery pipeline presented here may find general application for the discovery and preparation of oligoclonal mixtures of cross-neutralizing $V_H$Hs against other protein targets involved in human diseases. We foresee that such antibody mixtures could find utility against bacterial, viral, and parasitic infections, where multiple virulence factors or targets must be neutralized. Conversely, oligoclonal mixtures of highly specific antibodies could also be applicable for disease targets where cross-reactivity of single antibodies could be detrimental due to high homology between the target antigen and endogenous non-target antigens in areas such as oncology and autoimmunity. In conclusion, our work demonstrates the feasibility of using oligoclonal $V_H$H mixtures to neutralize complex snake venoms and shows the potential of discovering para-specific $V_H$Hs originally raised against toxins from other snake genera. Moreover, the discovery pipeline presented here can potentially be applied to address a range of other human diseases.

## Methods

All animals and in vivo methodologies used in the present work were approved by the bioethics committee of the IBt-UNAM under project # 385 "Caracterización funcional y análisis de especificidad de venenos de coralillos Norteamericanos". The Bioethics committee of the *Instituto de Biotecnología, Universidad Nacional Autónoma de México* (IBt-UNAM) is in compliance with the EU Directive 2010/63/EU for animal experiments[54].

### $V_H$H phage display library generation

Immunization followed by generation of a $V_H$H-displaying phage library targeting elapid snake venoms was commercially performed at the VIB nanobody core (Brussels, Belgium). For this, one alpaca and one llama were immunized with a mixture of venoms from 18 snake species (*Dendroaspis angusticeps, D. jamesoni, D. polylepis, D. viridis, Naja anchietae, N. annulifera, N. ashei, N. haje, N. katiensis, N.*

*melanoleuca, N. mossambica, N. nigricincta, N. nigricollis, N. nivea, N. nubiae, N. pallida, N. senegalensis,* and *Hemachatus haemachatus*), including Gerbu adjuvant P as an adjuvant. The composition of the venoms included for immunization can be found in a study by Nguyen et al.[55]. Both camelids were injected s.c. bi-weekly at eight time points with increasing doses of the respective venom mixtures (See Supplementary Table 1 for the immunization scheme). Venoms were mixed, diluted in phosphate-buffered saline (PBS: 137 mM NaCl, 3 mM KCl, 8 mM $Na_2HPO_4.2H_2O$, 1.4 mM $KH_2PO_4$, pH 7.4) and mixed with Gerbu adjuvant P before injection.

To construct the $V_H$H phage display libraries[56], peripheral blood mononuclear cells (PBMCs) were isolated from the blood samples collected on days 46, 49, 102, and 105. The isolated PBMCs were used for total RNA extraction and libraries were prepared by pooling the total RNA samples after 46 and 49 days to generate a first library and after 102 and 105 days to generate a second library. These two pools of the total RNA samples were used as templates for first-strand synthesis of cDNA using oligo(dT) primers. Thereafter, $V_H$H-encoding open reading frames were amplified by polymerase chain reaction (PCR), cloned into the phagemid vector pMECS, and transformed into electrocompetent *E. coli* TG1 cells.

### Camelid antibody titer determination by enzyme-linked immunosorbent assay (ELISA)

White 96-well Immuno Plates (GR-655074, Thermo Fisher Scientific) were coated with 60 μL/well of the different whole venoms diluted in PBS (0.5 μg/mL) and incubated overnight at 4 °C. The next day, the plates were washed 4 times with PBST (PBS + 0.1% Tween 20), blocked with 200 μL/well of 0.5% bovine serum albumin (BSA) in PBST for 1 h at room temperature (RT), and washed 4 times with PBST. Then, 60 μL/well of plasma samples diluted to 0.4% (v/v) in 0.5% BSA-PBST were added and incubated for 1 h at RT, followed by 4 washes with PBST. Bound IgGs were detected with 60 μL/well of HRP-conjugated anti-alpaca IgG $V_H$H domain (128-035-232, Jackson ImmunoResearch) diluted 1:10,000 in 0.5% BSA-PBST and incubated for 1 h at RT, followed by 4 washes with PBST. Finally, 60 μL/well of SuperSignal™ ELISA Pico Chemiluminescent Substrate (37070, Thermo Fisher Scientific) was added, and the plates were incubated for 5 min at RT before reading in a plate reader (VICTOR® Nivo™, PerkinElmer).

### Venoms and purification of toxins

Venom from *M. fulvius* was obtained as a pool from 67 individual specimens, kindly donated by Jack Facente from "AGRITOXINS Venom Lab" (Florida, US). Venom from *M. diastema* was manually extracted from a single specimen collected in Los Tuxtlas, Veracruz, Mexico (Collection license # SGPS/DGVS/03459/15, SEMARNAT, Mexico) and kept at the "Herpetario Cantil" of IBt-UNAM, Cuernavaca, Mexico. The extracted venom was recovered using milli-Q $H_2O$, centrifuged at 12,100 x g to remove cellular debris, lyophilized, and kept at 4 °C until use. Neurotoxins to be used as antigens for phage display campaigns were selected based on their abundance in the venoms of either *M. fulvius*[6] or *M. diastema*, their high lethality in rodent models, and their similarity to toxins present in other North American coral snake venoms[6,7,30,42,44,50,57,58]. The two main $PLA_2$ neurotoxins from *M. fulvius* venom ($PLA_2$N and $PLA_2$O), and the $PLA_2$-containing fractions from *Naja nigricollis* venom (Nn19), *Naja melanoleuca* (Nm15), and *Hemachatus haemachatus* (Hh3), together with short-chain αNTx-containing fractions from *M. diastema* (DH), *Naja haje* (Nh1), *Dendoaspis viridis* (Dv1), and *Hemachatus haemachatus* (Hh1) were purified from whole venoms using reversed-phase high-performance liquid chromatography (RP-HPLC) with a $C_{18}$ column using the gradient described previously[6,7,55]. In addition, α-cobratoxin from *Naja kaouthia* was purchased from Latoxan, and a $PLA_2$ from the venom of *Echis pyramidum* was purified through size-exclusion chromatography employing a Superdex 75 Increase 10/300 GL column (Cytiva) pre-equilibrated with

PBS. 5 mg/mL of venom diluted in PBS was added to the column and eluted at a flow rate of 0.5 mL/min in 500 μL fractions. Subsequently, these fractions underwent analysis via sodium dodecyl sulfate poly-acrylamide gel electrophoresis (SDS-PAGE) using a 10% Bis-Tris gel in MES buffer. Fractions displaying a band corresponding to the molecular weight of PLA$_2$s (~14 kDa) were pooled for further use.

## Expression and purification of recombinant toxins

Three short-chain αNTxs from the 3FTx family were used as target antigens for V$_H$H discovery. Two of them, eurytoxin and αNTx DH, have previously been identified in coral snake venoms[30,31], and the third one, scNTx, is a consensus protein, designed based on the sequences of eleven αNTxs from venoms of different elapids[32]. All three short-chain αNTxs were recombinantly produced in *E. coli*, using the SHuffle® T7 strain (New England Biolabs) for recombinant eurytoxin (rEury) and Origami Gold DE3 (Novagen®) for recombinant αNTx DH (rDH) and scNTx. Glycerol stocks of *E. coli* cells transformed with the pQE30 vector containing the toxins were kindly provided by Alejandro Olvera (rEury and rDH) and Dr Gerardo Corzo (scNTx) from IBt-UNAM. Cells from the glycerol stocks were used to inoculate 50 mL LB medium supplemented with ampicillin (80 μg/mL) and grown until an OD$_{600}$ of 0.7 was reached. Next, 10 mL of these cultures were used to inoculate 1 L of LB medium and protein expression was induced by addition of 0.1 mM IPTG. Thereafter, the cells were cultured for 24 h at 16 °C and 250 rpm for protein expression. The toxins were purified from the supernatants by gravity flow purification using immobilized metal-ion affinity chromatography (HIS-Select® Nickel Affinity Gel, Merck Millipore). The slurry was equilibrated with PBS before adding the supernatant, after which the slurry was washed with PBS and the toxins eluted using 250 mM imidazole. The imidazole was removed by dialysis against PBS (Spectra/Por® dialysis membrane 3.5 kDa MWCO). Further purification was achieved using RP-HPLC on a C$_{18}$ column equilibrated with 0.1% trifluoroacetic acid (TFA) and eluted using a gradient towards acetonitrile with 0.1% TFA[7]. All purified toxins were lyophilized and stored at 4 °C until use. The identity and integrity of the toxins were verified using mass spectrometry with an electrospray ionization system (ESI-MS) on an LTQ-Orbitrap XL mass spectrometer (Thermo Fisher Scientific).

## Biotinylation and mass spectrometry analysis of toxins

Lyophilized toxins were resuspended in PBS to a final concentration of approximately 2 mg/mL for biotinylation. The toxins were biotinylated using EZ-Link™ NHS-PEG$_4$-Biotin (A39259, Thermo Fisher Scientific) at a 1:2 molar ratio (2 biotin for every toxin molecule) for 30 min at RT. Free biotin was removed using 2000 Da MWCO filter tubes (Sartorius). The toxin concentrations were determined using absorbance at 280 nm (NanoDrop, Thermo Fisher Scientific) and calculated based on their molar extinction coefficients, which were obtained in silico using the Expasy ProtParam tool (https://web.expasy.org/protparam/).

The molecular mass of all the toxins[6,31] and the biotinylation ratio was determined by MALDI-TOF MS using an Ultraflex II TOF/TOF spectrometer (Bruker Daltonics).

## Phage display selection campaigns and subcloning

To select for toxin-binding V$_H$Hs, phage display selection campaigns were performed using the V$_H$H-displaying phage library[17,19]. In short, three consecutive rounds of selection were performed for three αNTxs (scNTx, rDH, and rEury) and two PLA$_2$s (PLA$_2$N and PLA$_2$O), incubating the phage library with final toxin concentrations of 50 nM in the first two rounds and 10 nM in the third round. For toxins rDH and rEury, round 3 (round 3b) was repeated using 50 nM antigen due to the low enrichments observed. After the third round, the V$_H$H-encoding genes were isolated from the glycerol stocks of the phage outputs, digested with PstI and Eco91I restriction endonucleases, subcloned into Xb-145 (a modified pHEN6 expression vector with an OmpA signal peptide and a C-terminal

3xFLAG and 6xHis-tag), and transformed into *E. coli* BL21 (DE3) cells[33]. Subsequently, individual V$_H$H clones were picked into 500 μL of 2xYT medium supplemented with kanamycin (50 μg/mL) and glucose (2%) in 96-deep well plates and grown O/N at 30 °C and 800 rpm.

## Screening of V$_H$Hs for antigen binding using DELFIA

For V$_H$H expression, 10 μL of each overnight culture was used to inoculate 1 mL of autoinduction medium[59] supplemented with kanamycin (50 μg/mL) in 96-deep-well plates, and the cultures were incubated O/N at 30 °C and 800 rpm. Thereafter, the plates were centrifuged for 10 min at 3000 x g, and the pellets were frozen at −20 °C O/N. The next day, the pellets were resuspended in 110 μL PBS, centrifuged for 10 min at 4500 x g, and the supernatants (the periplasmic fractions in which the majority of the V$_H$H products were expected to exist) were transferred to a 96-well plate and stored at −20 °C until use. The V$_H$Hs were screened for binding using an expression-normalized capture DELFIA[18], where a 1:100 dilution of V$_H$H-containing periplasmic fractions in 3% milk-PBS was added to the 96-well MaxiSorp plates (Thermo Fisher Scientific) coated with 2.5 μg/mL of anti-FLAG M2 antibody O/N (F3165, Sigma-Aldrich), followed by addition of 100 nM of biotinylated toxins rEury, rDH, or scNTx. The bound toxins were detected with 0.2 ng/μL of Europium-labelled streptavidin diluted in DELFIA assay buffer (Perkin Elmer), followed by addition of 100 μL DELFIA enhancement solution (Perkin Elmer) per well. Binding was assessed via measuring Time-Resolved Fluorescence (TRF) signal at 337 nm (excitation) and 615 nm (emission), using a plate reader (VICTOR® Nivo™, PerkinElmer).

The plasmids encoding V$_H$H binders with the highest signal against each target antigen were purified (GeneJET Plasmid MiniPrep kit, Thermo Fisher Scientific) according to the manufacturer's protocol and sequenced using the M13rev-29 primer (Eurofin Genomics). The V$_H$H frameworks and CDRs were annotated and analyzed to identify unique clones (CLC Main workbench v22.0.2).

## Expression and purification of V$_H$Hs

In total, six V$_H$Hs were selected for expression and purification: three from the PLA$_2$ selection campaigns (TPL0637_01_A01, TPL0637_01_A07, and TPL0638_01_C09) and three from the αNTx selection campaigns (TPL0629_01_D11, TPL0629_01_A07 and TPL0629_01_G06). V$_H$H expression was performed using *E. coli* BL21 (DE3) in TB medium supplemented with kanamycin (50 μg/mL), glucose (0.1% w/v), and MgSO$_4$ (1 mM). The cells were grown at 37 °C and 220 rpm until OD$_{600}$ reached 0.5. Subsequently, 0.5 mM IPTG was added to induce protein expression for 16 h at 30 °C and 220 rpm. The cells were collected by centrifugation at 4,000 x g for 15 min at 4 °C and stored at −20 °C. The frozen cells were then resuspended in cold PBS supplemented with 10 mM imidazole and an EDTA-free protease inhibitor cocktail (Roche). Next, the periplasmic fractions containing the His-tagged V$_H$Hs were collected by centrifugation at 20,000 x g for 45 min at 4 °C. The V$_H$Hs were then captured on an affinity resin by gravity flow (HIS-Select® Nickel Affinity Gel, Merck Millipore). Unbound proteins were washed away with wash buffer (PBS with 200 mM NaCl and 20 mM imidazole). The V$_H$Hs were then eluted from the resin (PBS with 200 mM NaCl and 250 mM imidazole), after which the imidazole was removed by dialyzing against PBS (SnakeSkin™ dialysis tubing, Thermo Fisher Scientific; 3.5 kDa MWCO). The purity of the V$_H$Hs was analyzed by SDS-PAGE and the V$_H$H concentration was determined by absorbance at 280 nm (NanoDrop, Thermo Fisher Scientific), and calculated by their molar extinction coefficients which were determined using the Expasy ProtParam Tool (https://web.expasy.org/protparam/).

## Binding analysis in dose-response DELFIA

To assess cross-reactivity of the V$_H$Hs, dose-response DELFIAs were performed as described for the screening of V$_H$Hs with a few

exceptions. These included adding 5 μg/mL of purified $V_H$Hs instead of $V_H$H-containing periplasmic fractions to the anti-FLAG coated plates. Also, instead of using a single toxin concentration, the targets were first diluted to 1 μM, and then titrated 1:3 in 10 consecutive dilution steps and added to the plate. All other steps were identical to those described earlier. The targets consisted of a purified $PLA_2$ from *M. fulvius* venom ($PLA_2$N), $PLA_2$-containing fractions from the venom of *H. haemachatus* (Hh3), *N. melanoleuca* (Nm15), *N. nigricollis* (Nn19), and *E. pyramidum*, a purified αNTx from the venom of *M. diastema* (αNTx DH), αNTx-containing fractions from the venoms of *N. haje* (Nh1), *D. viridis* (Dv1), and *H. haemachatus* (Hh1), and α-cobratoxin purchased from Latoxan.

### Binding analysis using bio-layer interferometry

Binding kinetics between the purified $V_H$Hs and their specific toxin targets were analyzed using bio-layer interferometry. Measurements were performed in kinetics buffer (PBS and 0.02% Tween 20; ForteBio) at 30 °C using an Octet RED 96 instrument (ForteBio). Biotinylated $PLA_2$N or αNTx DH with a final concentration of 1 μg/mL were loaded onto streptavidin biosensors (Sartorius) until a thickness of approximately 0.9 nm was reached. Toxin-loaded biosensors were dipped into five $V_H$H concentrations (200, 66.7, 22.2, 7.4, 2.5, and 0.8 nM). The association of $V_H$Hs to the toxins was measured for 600 s, followed by the dissociation for 600 s by incubating the biosensors in kinetics buffer. Sensors were regenerated by two rounds of 5 s incubations in Glycine at pH 1.5, followed by kinetics buffer before measuring the binding kinetics of the next $V_H$H-toxin pair. Two reference measurements, one without biotinylated toxin and the highest concentration of $V_H$H, and one with biotinylated toxin but without any $V_H$H, were subtracted from all curves. All data were analyzed using Octet® Analysis Studio 12.2.2.26 (ForteBio).

### In vitro neutralization of enzymatic $PLA_2$ activity

Determination of $PLA_2$ enzymatic activity was performed using the fluorometric EnzChek™ Phospholipase $A_2$ Assay Kit (Invitrogen) according to manufacturer's protocol. Fluorescence was measured using a plate reader (VICTOR® Nivo™, PerkinElmer) at an excitation wavelength of 480 nm and an emission wavelength of 530 nm. Measurements were made immediately after substrate addition and then every 30 s for 10 min to verify the linearity of the kinetics. The enzymatic activity was defined as the relative fluorescence obtained 5 min after substrate addition.

Neutralization of enzymatic activity was assessed by incubating 0.1 mg/mL of purified $PLA_2$s or $PLA_2$-containing fractions from different elapid snake venoms ($PLA_2$N from the venom of *M. fulvius*, Nn19 from *Naja nigricollis*, and Hh3 from *H, haemachatus*) with a 1:20 toxin to $V_H$H molar ratio of each anti-$PLA_2$ $V_H$H for 30 min at RT. $PLA_2$ activity was determined for each of the mixtures in duplicate. The $PLA_2$ activity in the presence of the $V_H$Hs was normalized by setting the activity of the toxin incubated with buffer only to 100%.

### In vitro neutralization of αNTx mediated blocking of nAChR activity (Automated Patch Clamp electrophysiology)

Automated planar whole-cell patch clamp experiments were conducted to evaluate the neutralizing capacity of the discovered $V_H$Hs on αNTx DH and scNTx-mediated blocking of nAChR activity. All electrophysiology experiments were performed using a human-derived Rhabdomyosarcoma RD cell line (American Type Culture Collection, ATCC), which endogenously expresses muscle-type nAChRs (α1, β1, δ, and γ-subunit), on a Qube 384 automated patch clamp platform (Sophion Bioscience) with 384-channel patch chips (patch hole resistance 2.00 ± 0.02 MΩ) as described elsewhere[19]. The nAChR-mediated currents were elicited by 70 μM acetylcholine (ACh) (corresponding to approximately the $EC_{80}$ concentration). A second ACh addition was used to evaluate the toxin effect in combination with each of the three

discovered anti αNTx $V_H$Hs at different concentrations (5, 15, 45, and 135 nM). The toxin concentration (αNTx DH = 15 nM; scNTx = 5 nM) was chosen to be approximately the previously determined $IC_{80}$ value, which is the toxin concentration that inhibits 80% of the maximum ACh current. Toxins and $V_H$Hs were preincubated for at least 30 min before addition to the cells and the patched cells were incubated with the toxin-$V_H$H mixtures for 5 min before the second ACh addition. The inhibitory effect of the toxins on the elicited ACh current was normalized to the full ACh response and averaged in the group ($n = 8$). The data was analyzed with Sophion Analyzer v6.6.70 (Sophion Bioscience) and GraphPad Prism v10.

### Design, expression, and purification of bivalent-$V_H$H and $V_H$H-Fc constructs

For expression of bivalent TPL0629_01_D11, the pUC57 vector containing $V_H$H-$(GGGGS)_3$-$V_H$H was purchased as a synthetic gene from GenScript. The plasmid was transformed into XL1-Blue cells (Agilent), amplified, and purified using a miniprep kit following the manufacturer's instruction (GeneJET Plasmid MiniPrep kit, Thermo Fisher Scientific). Afterwards, the purified plasmid was digested using *Not*I and *Pst*I restriction enzymes (New England Biolabs) and ligated into the Xb-145 expression vector using T4 DNA ligase (New England Biolabs). After transformation into BL21(DE3) cells, a positive transformant was used for the expression and purification of a bivalent TPL0629_01_D11 construct similarly as explained for monovalent $V_H$Hs.

For expression of $V_H$H-Fc, the nucleotide sequence of the constant heavy chain domain 2 and 3 from an human IgG1 antibody, harboring the LALA/YTE[60,61] mutations, was PCR amplified from the proprietary pINT3[19] vector and subjected to *EcoR*I and *Not*I digestion. Thereafter, the constant heavy chain domain 1 from the proprietary plasmid pINT12[19] was excised using the *EcoR*I and *Not*I restriction enzymes and replaced with the PCR amplified constant heavy chain domain 2 and 3.

The nucleotide sequence of the TPL0629_01_D11 $V_H$H was PCR amplified and integrated into the newly constructed vector using NEBuilder assembly, resulting in the generation of the expression plasmid TPL0629_01_D11-Fc (LALA, YTE). ExpiCHO cells (Thermo Fisher Scientific) were cultured and transfected according to the manufacturer's protocol using the plasmid at a concentration of 1 μg DNA/mL and ExpiFectamine. Transiently transfected cells were cultivated for four days at 125 rpm, 37 °C, 8% $CO_2$, and 70% humidity. Following incubation, cells were harvested, and $V_H$H-Fc in the supernatant was purified using affinity chromatography on a MabSelect SuRe column (Neo Biotech).

### In vivo neutralization experiments

All in vivo experiments were performed with groups of three CD1 mice between 18 and 20 g of total body weight and indistinct sex. All mice were provided by the animal facility of IBt-UNAM and were kept under 12 h light and dark cycles with food and water *ad libitum*, ambient temperature of 18–24 °C and approximately 60% relative humidity. For neutralization of whole coral snake venoms, two coral snake species (i.e., *M. fulvius* and *M. diastema*), whose venoms contain identical or very similar toxins to the ones used as antigens in the phage display selection campaigns, were chosen. Toxin to $V_H$H molar ratios were calculated based on the approximate abundance of $PLA_2$s and 3FTxs in the venoms, obtained from proteomic data[6]. The number of 3FTx or $PLA_2$ molecules present in 3 $LD_{50}$s of venom was then used to calculate the necessary amount of $V_H$Hs in the mixture (Supplementary Table 4).

**Determination of $LD_{50}$s for purified toxins and whole venoms.** $LD_{50}$s were determined for $PLA_2$N (*M. fulvius*) and αNTx DH (*M. diastema*) toxins and the whole venoms of *M. fulvius* and *M. diastema* using the i.v. and s.c. routes. Groups of three mice were injected with varying doses of the toxin or whole venom in a final volume of 500 μL PBS for

i.v. and 100 μL PBS for s.c. injection. The survival percentage was determined 24 h after injection, and the data was analyzed using a non-linear regression (semi-logarithmic dose-response curve)[62].

**Preincubation experiments.** For preincubation experiments, 3 $LD_{50}$s of each toxin or venom were combined with their corresponding $V_HH$ or $V_HH$ mixture using a range of toxin to $V_HH$ molar ratios going from 1:1 to 1:10 in a total volume of 500 μL of PBS (Supplementary Tables 2 and 5), following the guidelines of the Mexican Pharmacopeia (9$^{th}$ Edition)[63]. Due to a revision in the ethical guidelines and protocols, the injection volume was decreased to 250 μL during the whole venom neutralization experiments. For comparison, the commercial polyclonal antivenom, Coralmyn, which is composed of purified equine F(ab')$_2$ fragments and is currently the only treatment available in Mexico for coral snake envenomation, was used. The antivenom was combined with either the venom of *M. fulvius* or *M. diastema* at an approximate venom to antivenom molar ratio of 1:5 (Supplementary Table 5). One vial of Coralmyn (Batch # B-2H-12) was resuspended in 1 mL of injectable saline (provided by the manufacturer) and protein concentration was determined by measuring absorbance at 280 nm and corrected using an estimated extinction coefficient for F(ab')$_2$ of 1.44. To calculate venom to Coralmyn molar ratio, 100% of the protein content of Coralmyn was assumed to be F(ab')$_2$. The blends were preincubated at 37 °C for 30 min and injected into groups of 3 mice using the i.v. route. The mice were observed during the first 3 h and then approximately every 6 h for appearance of envenomation signs. The percentage of survival was calculated up to 24 h after the injection.

**Rescue experiments.** Rescue experiments were designed to better represent real envenomation, where the toxin is injected first and then the therapeutic molecule is administered using the i.v. route. In these experiments, mice were envenomed using the s.c. route with 3 $LD_{50}$s of each toxin in a final volume of 100 μL PBS. Immediately after toxin injection, the corresponding $V_HH$ or $V_HH$ construct was injected using the i.v. route in a total volume of 500 μL PBS (Supplementary Table 3). The experiments were performed using a range of toxin to $V_HH$ molar ratios going from 1:1 to 1:10. Mice were observed during the first 3 h and then approximately every 6 h for the appearance of envenomation signs. The percentage of survival was calculated up to 24 h after the injection.

**Statistical analysis.** To estimate the significance of the results obtained in the in vivo experiments, we performed a Mantel-Cox log-rank test[64]. Data were compared either to the negative control (PBS only) or to the commercial antivenom, Coralmyn. The significance value was set to α = 0.05, and therefore P-values higher than this were considered as non-significant.

### Reporting summary
Further information on research design is available in the Nature Portfolio Reporting Summary linked to this article.

## Data availability
All the data supporting the present manuscript is available in the form of Source Data Files and in the supplementary material. Relevant $V_HH$ and toxin sequences as well as detailed information on in vivo experiments are provided in the Supplementary Material. Raw data and analyses performed for the figures are available as Source Data Files. Source data are provided with this paper.

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

## Acknowledgements

The V$_H$H expression vector was a kind gift from the laboratory of Professor Bart De Strooper at the KU Leuven and VIB. The authors sincerely thank Alejandro Olvera and Dr. Gerardo Corzo for kindly providing the plasmids for expression of the αNTxs and Manuel Yañez Mendoza for his assistance during these expressions. We also thank Dr. Edgar Neri-Castro and Vanesa Gómez-Zarzosa for their assistance in coral snake handling and venom extraction. The authors also thank Jack Facente for kindly providing the pool of *M. fulvius* venom used for the present research as well as Lorenzo Seneci and Mathieu Roumet for proof-reading of the present manuscript. Finally, the authors thank Sergio González and Dr. Elizabeth Mata from the animal facility at IBt, UNAM, for mice reproduction and wellbeing management, as well as training regarding good practices for in vivo experiments. AHL is supported by a grant from the European Research Council (ERC) under the European Union's Horizon 2020 research and innovation program [850974], a grant from the Villum Foundation [00025302], and a grant from Wellcome [221702/Z/20/Z]. TPJ has received funding from the European Union's Horizon 2020 research and innovation program under the Marie Sklodowska-Curie grant agreement [713683] (COFUNDfellowsDTU). MBV is supported by a Eurotech Postdoctoral fellowship from the European Union's Horizon 2020 research and innovation programme under the Marie Skłodowska-Curie grant agreement [899987]. AA has received funding from the Mexican *Consejo Nacional de Ciencia y Tecnología*, FORDECyT-PRONAII [303045]. SS, SPB, and BGV are supported by a grant from the Novo Nordisk Foundation [NNF20SA0066621].

## Author contributions

Conceptualization: M.B.V., A.H.L. Methodology: M.B.V., A.L., Y.W., G.T.N., T.W.E., A.G.M., G.R.B., K.B., T.P.J., S.A., S.S., S.P.B., A.H.L. Investigation: M.B.V., A.L., Y.W., G.T.N., T.W.E., F.J., H.E., A.G.M., G.R.B., K.B., S.A., C.H.D., S.S., S.P.B. Visualization: M.B.V., A.L., Y.W. Funding acquisition: A.A., M.B.V., T.P.J., B.G.V., A.H.L. Project administration: A.L., M.B.V., A.H.L. Resources: A.A., A.H.L. Supervision: A.L., A.H.L. Writing – original draft: M.B.V., A.L., Y.W., T.P.J. Writing – review & editing: M.B.V., A.L., A.H.L., T.P.J., A.A., S.A., A.H.L.

## Competing interests

The authors declare the following competing interests: M.B.V., A.L., and A.H.L. are inventors on a submitted patent application (EP23192644.5), owned by the Technical University of Denmark. The remaining authors declare that they have no competing interests.
