## [Peer Review File · Nature Communications]

In vivo neutralization of coral snake venoms with an oligoclonal nanobody mixture in a murine challenge modelREVIEWER COMMENTS

Reviewer #1 (Remarks to the Author):

Major comments

The theme of this manuscript is centred around use of oligoclonal VHH mixtures for broad neutralization of snake venoms. I was asked specifically to comment on the antibody discovery and therapeutics/phage display aspects. I would characterize these aspects as 'standard, conventional and routine.' I don't mean this in a negative way, the approach is scientifically and technically sound. The authors immunized a llama and an alpaca with a mixture of 18 snake venoms and then selected binding VHHs by panning against a subset of the targets immunized against: two wildtype PLA2s purified from the venom of *Micrurus fulvius* (PLA2N and PLA2O), two recombinantly expressed native α NTxs (rEury 124 and rDH), and a recombinantly expressed short-chain consensus α NTx (scNTx).

Overall it is not clear whether the multiplexed immunization was helpful or not in generating cross-reactive VHHs. Likewise, the use of the consensus α NTx -- which was previously described -- may or may not have been helpful, although notably binding and neutralization data using this protein are not expected to be biologically relevant due to its artificial nature. Some of the key claims and hypotheses of the MS appear not to have been evaluated. The term 'broad neutralization' is used but breadth of binding or neutralization of individual VHHs or VHH cocktails is not analyzed. The notion that an oligoclonal VHH mixture is more effective in detoxification than individual VHHs is supported by the synergistic activity of two VHHs targeting different venom toxins (PLA2 and α NT); however, whether this 2-antibody cocktail is an 'optimal' combination or superior to other mAbs or pAbs is unclear since there is no comparison. Similarly, it is suggested that the molar ratio of VHHs in the oligoclonal mixture is designed based on the expected amounts of toxins to be present but no data are presented to support the idea that this was superior to other ratios.

In a broader sense I understand there have been multiple Abs previously developed that neutralize PLA2 and α NT, so I did not understand what was fundamentally different here that would make this study more impactful. If these particular antibodies are superior that is not clear from the data (no comparison). I assume that synergy between mAb neutralization of PLA2 and α NT (the 2 main components of elapid venom) has been previously demonstrated.

Minor comments

1. P1L21 the concept of neutralization or 'broad neutralization' is not relevant to cancer, or to many bacterial infections.
2. P2L33-38 I hope these drawbacks are raised again when talking about the VHH cocktail. Why not mix pAb antivenoms against several species? Batch-to-batch variation is a potential issue although not insurmountable, and 'low content of therapeutically active antibodies' is more of a dosing consideration.
4. P2L59 replacing pAbs with recombinant Abs - this is an idea associated with the 'antibody reproducibility crisis' and generally accurate but the relevance is very much less clear for pAb therapies. We are not talking about reproducibility, but clinical reliability/effectiveness and there are cost considerations.

5. Fig. 1, methods suggest this is a standard ELISA which seems inconsistent with the data format (relative light units). This is difficult to interpret due to lack of preimmune sera and lack of titration. Also no negative control protein tested and I think this is total IgG, not heavy chain only IgG.

6. Fig. 2, some biphasic dissociation for the 2-3 highest affinity VHHs, according to methods SEC was not done prior to BLI so could be aggregates present. Why wasn't ELISA/BLI done and presented for all the venoms/toxins, since they are available? Highly relevant to the idea of 'broad neutralization.'

7. Fig. 3, positive control benchmarks for reference and negative controls not done. Not sure neutralizing of two PLA2 preparations is strong evidence of 'broad neutralization.' Also not sure how closely related the PLA2 protein sequences are. The fact that all antibodies tested cross-neutralize may tell you something about how much of a special accomplishment this is.

8. Fig. 4, positive control benchmarks for reference and negative controls not done. I suppose if detoxification is happening it must occur very quickly since VHH monomers have a half-life of about 20 mins in rodents. Would you expect that the VHH-Fc fusions would be much more potent?

9. The sequences of the VHHs are not given which might conflict with NPJ reporting guidelines.

Reviewer #2 (Remarks to the Author):

The genesis of the submission is based on application of an immune VHH library derived from an alpaca and llama immunized with a combination of 18 elapid venoms. The concept focuses on the idea of generating a hyperimmune state in the host animal against multiple venoms to yield cross specific antibodies against venoms from different genera. The work was designed for coral snake envenoming. The library was screened for a set of VHHs and pooled in an oligoclonal mixture for validation.

Comments:

The total venom on day 0 does not add up to 18 venoms. As the single venom was 0.02 mg but the total venom was only 0.27 mg. Is this just a mistake or was there any venom excluded on day 0.

The data of antibody clustering can be added to the supplementary or text.

Can the authors provide the DELFIA results so the justification of the 2 sets of 3 clones that was further analyzed can be supported.

A key point that could be considered a limitation of the proposed solution is that the variation of the target antigens and the use of venom from cobras and mambas instead of the coral snake could be the key factor resulting in lower affinity clones being identified. This would render the approach for as a broad based solution lacking. Perhaps the authors could

provide a proposed solution to improve the affinity of the clones as a solution to this.

The authors used whole venom for immunization and screened against NTx and PLA. Though the justification for using these two toxins were mentioned in text, does the author have any information about the % of abundance of the toxins in the venom? This would have some influence on the immune response of the host animal to an extent and relate to the lower affinity of the clones. This could be mentioned in the introduction or discussion if the information is available.

Reviewer #3 (Remarks to the Author):

The authors have presented an interesting work on the discovery and development of broadly neutralising VHH oligo mixtures. However, the following concerns need to be addressed before the manuscript can be accepted for publication.

Major comments

1. While the immunisation was performed against a mixture of venoms from 18 elapid snakes, and ELISA was performed with these venoms to demonstrate seroconversion, why were the VHH tested only against coral snake venoms? As the authors mention that there is a high sequence similarity between *Naja* and *Dendroaspis* toxins (Line 118-120), why didn't the authors check for cross-neutralisation against these snakes? Though this study had the potential to demonstrate the broadly neutralising capabilities of the VHH, the authors have restricted themselves to testing only against coral snake venoms. Evaluating neutralisation against other snake venoms rich in three-finger toxins and phospholipase A2s would significantly improve the impact of this paper.
2. Line 28 mentions that the coral snake venoms are predominantly composed of neurotoxic PLA2s and 3FTx. However, Line 40 mentions a low abundance of these toxins in these venoms. If the latter was accurate, what is the composition of these venoms? The authors could provide a brief note on the venom composition of these snakes.
3. Negative control is not shown in Figure 1. Have the authors assessed the level of background binding from naive sera? As the sera is only diluted to 0.4% v/v, it is important to show non-specific background binding at this dilution.
4. 2.3 BLI - Given that this step is used to down select binders, what was the replicability of this assay? Have the authors observed any change in pattern when the assay was repeated? How many replicates were used in this experiment? All of this must be clarified in the manuscript.
5. 2.4 Neutralisation of PLA2 activity - The authors conclude that the VHH are broadly neutralising by testing the VHH against only two PLA2s. Given the high-throughput nature of this assay, I strongly recommend including multiple group I and II PLA2s in this assay to represent the entire diversity of this toxin family. The authors could also include an unrelated PLA2 negative control in this assay to support the specificity of the VHHs.
6. The legends in Figures 3B and 3C are hard to follow. Clarity on response with toxin alone

and in the presence of VHH at different molar ratios mentioned in the results should be depicted in the figures for clarity.

7. Could the authors provide the dosage of VHH in mg/kg and the neutralisation potency in mg of toxin neutralised per ml of VHH to compare with the conventional products?

8. It would help the readers if the authors clarify which antivenoms are used for treatment coral snake envenoming. The authors could compare their VHH to this to show how they perform in vivo.

Minor comments

1. The amounts of adjuvants used is missing from the methods section.

2. In which facility immunisation was performed?

3. What were the RIN values of the RNA isolates?

4. Statistical comparisons for statements, like 'lower binding signals were generally observed against Dendroaspis venoms compared to Naja venoms' are missing.

5. The legend for the figure 2 isn't clear. What do the colours indicate?

6. Lines no 123 and 124, provide clarity on the PLA2O, PLA2N, rEury and rDH abbreviations used here in the first mention.

7. Explain the process of DELFIA in the methodology briefly.

8. Could the authors provide any reference to highlight that they have not exceeded the maximum permissible i.v injection volume?

We have now addressed all the reviewers' comments, please find them below. We thank the editor and the reviewers for their contribution to the improvement of our work.

REVIEWER COMMENTS

Reviewer #1 (Remarks to the Author):

Major comments

The theme of this manuscript is centred around use of oligoclonal VHH mixtures for broad neutralization of snake venoms. I was asked specifically to comment on the antibody discovery and therapeutics/phage display aspects. I would characterize these aspects as 'standard, conventional and routine.' I don't mean this in a negative way, the approach is scientifically and technically sound. The authors immunized a llama and an alpaca with a mixture of 18 snake venoms and then selected binding VHHs by panning against a subset of the targets immunized against: two wildtype PLA2s purified from the venom of *Micrurus fulvius* (PLA2N and PLA2O), two recombinantly expressed native α NTxs (rEury 124 and rDH), and a recombinantly expressed short-chain consensus α NTx (scNTx).

We thank the reviewer for the valuable comments on our manuscript, which we believe contributed to significantly improving our work.

Overall it is not clear whether the multiplexed immunization was helpful or not in generating cross-reactive VHHs. Likewise, the use of the consensus α NTx -- which was previously described -- may or may not have been helpful, although notably binding and neutralization data using this protein are not expected to be biologically relevant due to its artificial nature. Some of the key claims and hypotheses of the MS appear not to have been evaluated.

We agree with the reviewer's remark that the significance of using such a diverse venom mixture for immunization was not evaluated. The main objective of the generation of the library was to provide a resource that could be used for discovery of nanobodies against toxins in the venoms from multiple snake species. Therefore, we selected a combination of venoms for immunization that could provide a wide coverage, focusing mainly on medically relevant snake species from sub-Saharan Africa. Previous studies have proven the utility of complex immunogen mixtures for immunization in the generation of polyclonal sera with broad neutralization capacities (e.g., <https://doi.org/10.1371/journal.pntd.0007250>). In agreement with this, the discovery of nanobodies that can recognize and neutralize toxins from venoms, both present and not present in the immunization mixture, suggests that the strategy was successful, although a specific comparison of this nanobody library with others, generated using llamas immunized with individual venoms, is outside the scope of the present work. We have therefore avoided claiming that the immunization strategy is better than other, but simply just stated that it works, which we believe is supported by the data.

Regarding the use of the recombinant consensus toxin scNTx, phage display campaigns were performed both with this and with two recombinant native toxins from coral snake venoms in parallel. However, the titers of the phage pool outputs showed a much higher enrichment using the consensus scNTx compared to the native toxins. Despite the low enrichment on native toxins, a limited number of clones from the selection outputs on native toxins were screened for binding to their respective antigen. However, only a few clones showed any binding with low binding signals (indicating low affinity). In contrast, screening of nanobodies from the selection campaign utilizing

the recombinant consensus scNTx resulted in multiple nanobodies (with a high sequence diversity) binding with high binding signals when analyzed using an expression normalized DELFIA. We have added these data to the supplementary material (Supplementary Figure 3) and mentioned the differences in lines 137-146.

The term 'broad neutralization' is used but breadth of binding or neutralization of individual VHHs or VHH cocktails is not analyzed

Regarding the cross-binding claim, we thank the reviewer for pointing this out. We have added data demonstrating that the selected nanobodies can recognize toxins from different elapid genera and incorporated it to the manuscript in lines 158-169, 361-367, and in Supplementary Figures 5 and 6. Here, we show that the anti-PLA₂ V_HHs are able to recognize PLA₂s from the elapid genera *Naja* and *Haemachatus* but are not able to recognize a PLA₂ from the viperid snake *Echis pyramidum*. Similarly, we show that the anti-3FTx V_HHs can recognize short chain neurotoxins from three different elapid genera: *Naja*, *Haemachatus*, and *Dendroaspis*, in addition to the coral snake toxins, but not the long chain neurotoxin, alpha-cobratoxin from *N. kaouthia*. The mentioned toxins were identified based on their elution time using RP-HPLC and their molecular weight determined by MALDI-TOF MS. To evaluate how the observed cross-binding correlates to broad-neutralization, inhibition of PLA₂ from some of these snake species was evaluated and added to lines 195-199, 365-369.

The notion that an oligoclonal VHH mixture is more effective in detoxification than individual VHHs is supported by the synergistic activity of two VHHs targeting different venom toxins (PLA₂ and aNT); however, whether this 2-antibody cocktail is an 'optimal' combination or superior to other mAbs or pAbs is unclear since there is no comparison.

A comparison with Coralmyx, the antivenom used in Mexico for the treatment of coral snake envenomation is now included. This showed that the commercial antivenom neutralizes the venom of *M. fulvius* at similar doses to the tested oligoclonal mixture. Conversely, the venom of *M. diastema*, partially neutralized by the oligoclonal mixture, is not neutralized by Coralmyx at the selected dose. These data demonstrate that the oligoclonal mixture tested here has similar, or possibly better neutralization properties than the current commercial antivenom; this has been added to lines 308-312 and to Figure 6. We further show that these two V_HHs are both necessary and sufficient, to neutralize whole venom-induced lethality, while the individual V_HHs are unable to do so on their own (Figure 6).

Similarly, it is suggested that the molar ratio of VHHs in the oligoclonal mixture is designed based on the expected amounts of toxins to be present but no data are presented to support the idea that this was superior to other ratios.

At this stage, we do not know or claim that this oligoclonal mixture is the optimal combination of V_HHs but rather the minimal mixture that is sufficient and necessary to neutralize the lethality of complete coral snake venom(s). We have now clarified this in line 297.

To provide more detailed data regarding the rationale behind the design of the oligoclonal mixture, we have now added the RP-HPLC profile of *Micrurus fulvius* venom showing the abundance of different toxin families (supplementary Figure 8) and added detailed calculations on how the oligoclonal mixture was designed to match the venom compositions (Supplementary Table 4).

In a broader sense I understand there have been multiple Abs previously developed that neutralize PLA₂ and aNT, so I did not understand what was fundamentally different here that would make this study more impactful. If these particular antibodies are superior that is not clear from the data (no comparison). I assume that synergy between mAb neutralization of PLA₂ and aNT (the 2 main

components of elapid venom) has been previously demonstrated.

In previous studies, our group showed that it was possible to neutralize the lethality of a whole venom using human monoclonal IgG antibodies, where lethality was mainly caused by a single toxin. This new work represents the first time that a monoclonal antibody mixture can neutralize the lethality of more complex venoms where the clinical syndrome is caused by more than one toxin class. Furthermore, this is the first time an oligoclonal mixture of V_HHs is used instead of IgGs.

There is no described synergy between mAbs targeting PLA₂s and 3FTxs, but previous studies of coral snake venom composition and toxicity have shown that various PLA₂s and αNTxs contribute to the overall lethality of a single venom and are enough on their own to cause lethality. In current antivenoms, neutralization of these different toxins is achieved through generation of polyclonal sera through hyperimmunization. Therefore, the neutralization of a complete venom with an oligoclonal mixture of only two mAb is expected to be viewed as both a surprising and highly relevant achievement in the field of toxinology that opens the door for the design of other oligoclonal mixtures against even more complex venoms.

In order to provide a relevant comparison to the current treatment, we have now added neutralization assays with the antivenom Coralmyn, a polyclonal F(ab')₂ antivenom that is the only available therapeutic in Mexico for treatment of coral snake envenomation (Figure 6) and lines 308-312.

Minor comments

1. P1L21 the concept of neutralization or 'broad neutralization' is not relevant to cancer, or to many bacterial infections.

Broad neutralization can be an advantage for example if there are various mutated versions giving rise to similar but not identical proteins in many infectious diseases (e.g., broad neutralization of different toxinotypes from *Clostridium difficile*, *Vibrio cholera*, and various *E. coli*/*Shigella* infections). This has been clarified in line 21.

2. P2L33-38 I hope these drawbacks are raised again when talking about the VHH cocktail. Why not mix pAb antivenoms against several species? Batch-to-batch variation is a potential issue although not insurmountable, and 'low content of therapeutically active antibodies' is more of a dosing consideration.

Due to the heterologous nature of the antivenoms and their high price, mixing several antivenoms and/or giving higher doses increases the risk of adverse reactions and the price of the treatment. Therefore, there is not a single antivenom manufacturer that does this worldwide. We have clarified this in lines 42-44.

4. P2L59 replacing pAbs with recombinant Abs - this is an idea associated with the 'antibody reproducibility crisis' and generally accurate but the relevance is very much less clear for pAb therapies. We are not talking about reproducibility, but clinical reliability/effectiveness and there are cost considerations.

In snakebite envenomation, there is also the difficulty to neutralize all relevant toxins in the venoms due to their low immunogenicity and/or abundance. This has been clarified in lines 37-42 and a reference to price estimations on recombinant antivenoms is included in line 59.

It is well-recognized in the field that existing antivenoms are suboptimal (see e.g., Gutiérrez et al. Snakebite envenoming. Nature Reviews Disease Primers 2017, considered the authoritative review paper in the field).

5. Fig. 1, methods suggest this is a standard ELISA which seems inconsistent with the data format (relative light units). This is difficult to interpret due to lack of preimmune sera and lack of titration. Also no negative control protein tested and I think this is total IgG, not heavy chain only IgG. The assay in Figure 1 is indeed a standard ELISA with a chemiluminescence readout as described in Section 4.2 and, therefore, using the RLU data format. Day 0 in the figure is a preimmune sera, this has been clarified in the figure legend. Given the high number of venoms that needed to be tested, we did not determine a titer for all the time points against all venoms. Still, sera from a few timepoints were titrated (data not shown) to determine an optimal dilution to be used (0.4% which is added in line 462). This dilution was thereafter used to assess binding at all different time points. No negative antigen was included, however, the difference between the various venoms and the difference in binding signal at different timepoints indicate that an increased specific binding to the venoms is observed over time. Finally, it is indeed the total IgG that was detected, this has been corrected in line 463.

6. Fig. 2, some biphasic dissociation for the 2-3 highest affinity VHHs, according to methods SEC was not done prior to BLI so could be aggregates present. Why wasn't ELISA/BLI done and presented for all the venoms/toxins, since they are available? Highly relevant to the idea of 'broad neutralization.' No SEC was performed before the BLI experiments since we typically do not observe significant aggregation of V_HHs, although some extent of aggregation cannot be ruled out. Dose-response DELFIA assays have now been added to show the V_HH's broad binding to PLA₂s and αNTxs from various elapid genera (Supplementary Figures 5 and 6). In addition, the neutralization of PLA₂s from various elapid species have been added in figure 3A to demonstrate broad neutralization.

7. Fig. 3, positive control benchmarks for reference and negative controls not done. Not sure neutralizing of two PLA₂ preparations is strong evidence of 'broad neutralization.' Also not sure how closely related the PLA₂ protein sequences are. The fact that all antibodies tested cross-neutralize may tell you something about how much of a special accomplishment this is. We thank the reviewer for their suggestion and have expanded our sampling of PLA₂s in the *in vitro* neutralization assay, adding them to Figure 3. Here, we demonstrate neutralization of PLA₂s from the genus *Naja* but not from the genus *Hemachatus*. From our previous work (see e.g., Ledsgaard et al. mAbs 2022, Sørensen et al. Toxicon 2023, and Sørensen et al. Scientific Reports 2023), we are quite sure that it is not easy to achieve broad-neutralization of different toxins from the same family, as the sequence similarity may be below 40%. The V_HHs without toxin were added as negative controls and, as expected, no PLA₂ activity was observed.

8. Fig. 4, positive control benchmarks for reference and negative controls not done. I suppose if detoxification is happening it must occur very quickly since VHH monomers have a half-life of about 20 mins in rodents. Would you expect that the VHH-Fc fusions would be much more potent? We have now included Coralmyx, a polyclonal antivenom, as a positive control. This is the commercial antivenom used to treat coral snake envenomation in Mexico and is composed of purified equine F(ab')₂ fragments. We performed a comparative assay where we tested the neutralizing capacity of Coralmyx at an approximate 1:5 molar ratio (Figure 6), which due to their

bivalent nature, is comparable to the 1:10 molar ratio of the oligoclonal mixtures used in this study. At this dose, equivalent to approximately 40 mg/kg of Coralmyx, the venom of *M. fulvius* was properly neutralized by the antivenom, saving all the envenomated mice. In contrast, the venom of *M. diastema* was not neutralized and all envenomated mice died within the first 3 hours. This shows that, compared to the current treatment, the tested oligoclonal mixture is better at neutralizing the venom of *M. diastema* and roughly similar at neutralizing the venom of *M. fulvius*. This has been added to Figure 6, lines 308-312.

The most potent V_HH targeting αNTxs has now also been tested in two additional formats: V_HH-Fc and as a bivalent V_HH construct (Figure 5) showing that these constructs are not more potent than the normal V_HH in this case. This information has been added in lines 268-281 and M&M section 4.13.

9. The sequences of the VHHs are not given which might conflict with NPJ reporting guidelines. We have added the sequences of the discovered V_HHs in Supplementary Figure 4.

Reviewer #2 (Remarks to the Author):

The genesis of the submission is based on application of an immune VHH library derived from an alpaca and llama immunized with a combination of 18 elapid venoms. The concept focuses on the idea of generating a hyperimmune state in the host animal against multiple venoms to yield cross specific antibodies against venoms from different genera. The work was designed for coral snake envenoming. The library was screened for a set of VHHs and pooled in an oligoclonal mixture for validation.

Comments:

The total venom on day 0 does not add up to 18 venoms. As the single venom was 0.02 mg but the total venom was only 0.27 mg. Is this just a mistake or was there any venom excluded on day 0. We thank the reviewer for noticing this mistake which has now been corrected in Supplementary Table 1. (0.015 mg per venom was used)

The data of antibody clustering can be added to the supplementary or text. The number of unique clones and how many families they belong to is now included in lines 152-154.

Can the authors provide the DELFIA results so the justification of the 2 sets of 3 clones that was further analyzed can be supported.

We have now included data regarding the enrichment obtained during the phage display selection campaigns, as well as the results from the primary screening of monoclonal V_HHs in expression normalized monoclonal DELFIAs in lines 137-154 and Supplementary Figures 3 and 4. Since this assay is normalized for variations in expression levels between the V_HHs, the signal intensity mainly correlates to the affinity of the screened V_HHs. The clones with the highest signals were selected for further analysis.

A key point that could be considered a limitation of the proposed solution is that the variation of the target antigens and the use of venom from cobras and mambas instead of the coral snake could be

the key factor resulting in lower affinity clones being identified. This would render the approach for as a broad based solution lacking. Perhaps the authors could provide a proposed solution to improve the affinity of the clones as a solution to this.

In case the affinity of the discovered clones is too low we can for example 1) make new discovery campaigns including the antigens to which currently a too-low affinity is observed. 2) use *in silico* tools such as Bayesian optimization to improve the affinity and the cross-reactivity of the discovered clones. 3) Perform random mutagenesis of the CDR regions. 4) More time-consuming and costly, but potentially one could also perform new immunizations of camelids including the coral snake venoms for immunization, make new nanobody displaying phage libraries and repeat the discovery campaigns. Information about this has been added in lines 402-404.

The authors used whole venom for immunization and screened against NTx and PLA. Though the justification for using this two toxins were mentioned in text, does the author have any information about the % of abundance of the toxins in the venom? This would have some influence on the immune response of the host animal to an extent and relate to the lower affinity of the clones. This could be mentioned in the introduction or discussion if the information is available.

The protein composition of the venoms used for immunization can be found here: doi: 10.1093/gigascience/giac121, a note about this has been added in line 441. A chromatogram of whole venom from *M. fulvius* is added in Supplementary Figure 8 and information regarding toxin family abundance in both coral snake venoms is added to lines 292-297. As mentioned in lines 40-43, the immunogenicity of these toxins is rather low which may result in a low immune response upon immunization.

Reviewer #3 (Remarks to the Author):

The authors have presented an interesting work on the discovery and development of broadly neutralising VHH oligo mixtures. However, the following concerns need to be addressed before the manuscript can be accepted for publication.

Major comments

1. While the immunisation was performed against a mixture of venoms from 18 elapid snakes, and ELISA was performed with these venoms to demonstrate seroconversion, why were the VHH tested only against coral snake venoms? As the authors mention that there is a high sequence similarity between Naja and Dendroaspis toxins (Line 118-120), why didn't the authors check for cross-neutralisation against these snakes? Though this study had the potential to demonstrate the broadly neutralising capabilities of the VHH, the authors have restricted themselves to testing only against coral snake venoms. Evaluating neutralisation against other snake venoms rich in three-finger toxins and phospholipase A2s would significantly improve the impact of this paper.

In the present work, we focused on coral snake venoms because one of the main objectives was to test the utility of our phage display library for discovery of V_HHs against heterologous toxins (*i.e.*, toxins that were not part of the immunization mixture). In other projects that have not progressed as far, we are using the library to find V_HH against cobra and mamba toxins. Nonetheless, we agree with the reviewer that testing cross-binding of the discovered V_HHs with homologous elapid toxins is relevant for the present work as well. We have now included Supplementary Figures 5 and 6, where we show, using a dose response DELFIA, that the discovered V_HHs are able to recognize PLA2s from

Naja and *Haemachatus*, as well as 3FTxs from *Dendroaspis*, *Naja*, and *Haemachatus* and we have commented on this broad-recognizing properties in lines 165-169.

2. Line 28 mentions that the coral snake venoms are predominantly composed of neurotoxic PLA₂s and 3FTx. However, Line 40 mentions a low abundance of these toxins in these venoms. If the latter was accurate, what is the composition of these venoms? The authors could provide a brief note on the venom composition of these snakes.

Even though coral snake venoms are indeed composed primarily of PLA₂s and 3FTxs, not all these proteins are relevant for lethality in mammals, since some can be specific to their natural prey (mainly small snakes and lizards) or have functions that are not yet described. The 3FTxs that are highly lethal to mammals, known as alpha-neurotoxins (αNTxs), can be present in low amounts in some species and their small molecular weight usually makes them poorly immunogenic. We have tried to clarify this point in lines 25-31 and a note regarding specific venom composition considered for the design of the oligoclonal mixtures has been added in lines 368-373. A RP-HPLC chromatogram of *M. fulvius* venom has now been added in Supplementary Figure 8, for reference.

3. Negative control is not shown in Figure 1. Have the authors assessed the level of background binding from naive sera? As the sera is only diluted to 0.4% v/v, it is important to show non-specific background binding at this dilution.

The serum at day 0 was extracted before the injection of the first immunization and is therefore a preimmune serum. We have now clarified this in line 104 and in the legend of Figure 1.

4. 2.3 BLI - Given that this step is used to down select binders, what was the replicability of this assay? Have the authors observed any change in pattern when the assay was repeated? How many replicates were used in this experiment? All of this must be clarified in the manuscript.

We apologize for the confusion, BLI experiments were not used to downselect binders but instead to characterize the binders that were previously selected. The criteria for selection of binders was primarily a high signal in the expression normalized monoclonal DELFIA, which has been added as Supplementary Figures 2 and 3. This has now been clarified in lines 155-158.

5. 2.4 Neutralisation of PLA₂ activity - The authors conclude that the VHH are broadly neutralising by testing the VHH against only two PLA₂s. Given the high-throughput nature of this assay, I strongly recommend including multiple group I and II PLA₂s in this assay to represent the entire diversity of this toxin family. The authors could also include an unrelated PLA₂ negative control in this assay to support the specificity of the VHHs.

We thank the author for his suggestion, we have now included another elapid PLA₂ in the enzymatic assay (Figure 3A) and discuss the results in lines 195-201. Based on the results of the binding assays added in Supplementary Figure 5, we know that the V_HHs do not bind a PLA₂ containing fraction from viper venom (Group II), so this one was not included in the PLA₂ assay. With this data, we show that our V_HHs are able to neutralize at least one *Naja* and *Micrurus* PLA₂s, while they are not successful in neutralizing activity of a PLA₂ from *Hemachatus haemachatus*. These results show the specificity and broadly neutralizing activity of the discovered V_HHs.

6. The legends in Figures 3B and 3C are hard to follow. Clarity on response with toxin alone and in the presence of VHH at different molar ratios mentioned in the results should be depicted in the figures for clarity.

We have rephrased the legends and added the toxin only value to the graphs.

7. Could the authors provide the dosage of VHH in mg/kg and the neutralisation potency in mg of toxin neutralised per ml of VHH to compare with the conventional products?

Since only a few doses of V_HHs or the oligoclonal mixture were tested in the present work, a neutralization potency *sensu stricto*, where we determine the amount of V_HH that is able to save 50 or 100% of an envenomated mice population, was not performed. Nonetheless, we have now added the conversion of the dose used to mg/kg (lines 312-314) to facilitate the comparison with other data. We have also added a comparison with the current treatment (see below).

8. It would help the readers if the authors clarify which antivenoms are used for treatment coral snake envenoming. The authors could compare their VHH to this to show how they perform in vivo.

The commercial antivenom used to treat coral snake envenomation in Mexico is a polyclonal serum composed of F(ab')₂ fragments called Coralmyx. We have now performed a comparative assay where we test the neutralizing capacity of Coralmyx at an approximate 1:5 molar ratio (Figure 6). This is added in lines 308-314.

Minor comments

1. The amounts of adjuvants used is missing from the methods section.

The adjuvant has been added in line 592.

2. In which facility immunisation was performed?

We have now clarified that both immunization and the V_HH phage display libraries were generated at the VIB nanobody core in line 441.

3. What were the RIN values of the RNA isolates?

Since we did not construct the library ourselves, we do not have those details.

4. Statistical comparisons for statements, like 'lower binding signals were generally observed against Dendroaspis venoms compared to Naja venoms' are missing.

We did not perform any statistical analysis on this data and the mentioned statement is solely based on a visual inspection of Figure 1. We have now rephrased this to clarify in line 107.

5. The legend for the figure 2 isn't clear. What do the colours indicate?

The different colors are different V_HH concentrations, this has been added to the figure legend.

6. Lines no 123 and 124, provide clarity on the PLA2O, PLA2N, rEury and rDH abbreviations used here in the first mention.

These are the names given to the toxins when they were first described or expressed, we have now clarified this in lines 133-137.

7. Explain the process of DELFIA in the methodology briefly.

A brief description of the assay has been added in lines 549-553.

8. Could the authors provide any reference to highlight that they have not exceeded the maximum permissible i.v injection volume?

The injection volumes used were in accordance with the guidelines stated in the Mexican Pharmacopeia in its 9th edition, which states a specific 500 μ l volume for preincubation experiments (<http://www.farmacopea.org.mx>). A reference to these guidelines is added in line 690. Today, the guidelines have been revised and in order to adhere to the international standards, a lower injection volume (250 μ L) was used for the final experiments. All this is stated in M&M lines 690-692.

REVIEWER COMMENTS

Reviewer #1 (Remarks to the Author):

I was asked to comment on the antibody aspects of this paper and will again focus my comments on this. There is no argument that the authors have generated VHHs that bind and neutralize PLA2 and α NTx present in at least some venoms; however I believe neutralizing mAbs against these targets have been raised previously by other groups. I do not know how the VHHs compare with other mAbs or with pAb-based antivenoms in terms of binding/neutralization breadth.

Regarding the multiplex immunization, while I understand that the goal was to provide wide coverage, it remains possible that immunization with one or a few venoms could have been as or more successful. In their response the authors state that “previous studies have proven the utility of complex immunogen mixtures” but the paper cited in support of this statement involves immunization of horses with single venoms followed by mixing of monovalent pAb antivenoms?

Regarding the recombinant consensus toxin scNTx, failure to recover VHHs to two recombinant native toxins (rEury and rDH) is not very strong evidence that scNTx played an important role. Perhaps the better panning result with scNTx could be due to scNTx sharing greater homology than rDH in key regions with the scNTs within the venoms used for immunization, or at least those that were most immunogenic? There is no evidence that the scNTx was more useful than one or more recombinant native scNT(s) matched to the immunizing venoms – these experiments were not done.

Regarding cross-binding/breadth of neutralization, I defer to more qualified reviewers as to whether this is notable or not in the context of this field. Clearly binding is not always correlated to neutralization as shown by Hha3 PLA2 (similar binding to other PLA2s but no neutralization). It seems that the only confirmed example of neutralization breadth shown is PLA2N (*M. fulvius*) and Nn19 (*N. nigricollis*); and of DH (*M. diastema*) since this venom was not part of the immunization. It would have been nice to see binding (BLI) and in vitro neutralization data for the 18 venoms used for immunization and comparison with other ‘broadly neutralizing’ mAbs in the literature.

Regarding the comparison with Coralmyn, I defer to more qualified reviewers as to whether this is notable or not in the context of this field. I was uncertain whether *M. fulvius* and *M. diastema* venoms provide a full and fair comparison, since the VHHs were established during the workup to neutralize PLA2/scNT respectively from these species; Coralmyn is produced using *M. nigrocinctus* venom, which was not tested, and I’m not sure what is known regarding its neutralization breadth. The points I made previously stand (that other combinations of 2 or more VHHs, and different molar ratios, could be superior). It is not so clear to me that designing anti-venom mixtures based on component toxin abundance is automatically going to achieve the best result – and this hypothesis was not tested here.

Regarding the authors’ statements that “this new work represents the first time that a monoclonal antibody mixture can neutralize the lethality of more complex venoms where the clinical syndrome is caused by more than one toxin class” and “neutralization of a complete venom with an oligoclonal mixture of only two mAb is expected to be viewed as both a surprising and highly relevant achievement in the field of toxinology,” I defer to more qualified reviewers. This sounds like it could be a potentially useful contribution, but as a

non-expert in toxinology I assumed this had already been demonstrated, given that these 2 toxins are responsible for much of the clinical syndrome. I did not conduct an exhaustive literature search but synergy between pAbs against PLA2 and scNT (PMID 33423840) seems to be out there. If this is the major contribution of the paper it seems to me that this could be made more clear throughout, especially in the title and abstract.

Regarding minor point 7, it would be helpful to provide sequence alignments and % identity for all PLA2 and scNT used in the study. Don't PLA2 from *M. fulvius* and *N. nigricollis* share about 70% identity?

Regarding minor point 8, do the authors have an explanation as to why VHH monomer (15 kDa) and Fc fusion (80 kDa) behave similarly but bivalent VHH (30 kDa) is much less effective in Fig. 5? This seems rather odd.

Reviewer #2 (Remarks to the Author):

The authors have addressed all the points adequately.

Reviewer #3 (Remarks to the Author):

The authors have sufficiently addressed my queries. My only suggestion now is to add statistics to their survival curve figures.

We have now addressed the remaining reviewers' comments to the best of our ability, we thank the editor and the reviewers for their time and contribution to the improvement of our manuscript. Please find our answers below.

REVIEWER COMMENTS

Reviewer #1 (Remarks to the Author):

I was asked to comment on the antibody aspects of this paper and will again focus my comments on this. There is no argument that the authors have generated VHHs that bind and neutralize PLA2 and α NTx present in at least some venoms; however I believe neutralizing mAbs against these targets have been raised previously by other groups. I do not know how the VHHs compare with other mAbs or with pAb-based antivenoms in terms of binding/neutralization breadth.

Response:

We agree with the reviewer that mAbs have been previously generated against similar venom components, although, to the best of our knowledge, never particularly against coral snake venom toxins. So far, we have not compared our VHHs with other VHHs, since there are no relevant VHHs to compare with. However, we have compared the discovered VHH mixtures with Coralmyn, the current treatment for Coral snake envenomation, as highlighted in lines 326-332 and 407-411.

In this work, we demonstrate that a venom with more than one toxic component can be neutralized by mixing antibodies and, also, show that this is feasible to do with a smaller antibody format, such as VHHs compared to earlier work where IgGs have been used (see e.g., Khalek et al. *Science Translational Medicine* 2024 or Ledsgaard et al. *Nature Communications* 2023). Neutralization of a complete snake venom often requires that the VHHs cross-neutralize multiple similar toxins, since a single venom normally has more than one toxin of each subfamily. We have clarified this in lines 312 to 314.

The generation of polyclonal sera through animal immunization can allow for the generation of antibodies against many of these toxins but is not vis-à-vis comparable to our approach.

Regarding the multiplex immunization, while I understand that the goal was to provide wide coverage, it remains possible that immunization with one or a few venoms could have been as or more successful. In their response the authors state that “previous studies have proven the utility of complex immunogen mixtures” but the paper cited in support of this statement involves immunization of horses with single venoms followed by mixing of monovalent pAb antivenoms?

Response:

We believe it is relevant to mention the rationale behind using multiplex immunization but agree that we cannot conclude that it was better than immunizing with individual venoms. We have rephrased this in line 85-88.

The citation provided previously was meant to show an immunization strategy performed in coral snake venoms. We apologize for the lack of clarity of the previous response.

Due to the expensive and complicated nature of generating hyperimmune sera, direct comparisons between immunization strategies for the same group of snake venoms are seldom performed, although complex immunogen mixtures have been proven effective to

neutralize other elapid snake species (<https://doi.org/10.1371/journal.pntd.0004565>). Different immunization strategies and their pros and cons for production of polyclonal sera has been recently discussed in (<https://doi.org/10.3390/toxins15090517>).

Regarding the recombinant consensus toxin scNTx, failure to recover VHHs to two recombinant native toxins (rEury and rDH) is not very strong evidence that scNTx played an important role. Perhaps the better panning result with scNTx could be due to scNTx sharing greater homology than rDH in key regions with the scNTs within the venoms used for immunization, or at least those that were most immunogenic? There is no evidence that the scNTx was more useful than one or more recombinant native scNT(s) matched to the immunizing venoms – these experiments were not done.

Response: We agree with the reviewer that we have not thoroughly investigated if it is the consensus nature of scNTx that makes it a better phage display antigen in our experiment. We have rephrased to clarify this in lines 155-157. Nonetheless, our group has recently published a paper showing that the use of consensus toxins can be beneficial to discover more broadly-neutralizing antibodies (<https://onlinelibrary.wiley.com/doi/full/10.1002/pro.4901>), so we have reformulated our text to provide the rationale, but done our best to avoid saying that this approach is the most optimal one to use.

Regarding cross-binding/breadth of neutralization, I defer to more qualified reviewers as to whether this is notable or not in the context of this field. Clearly binding is not always correlated to neutralization as shown by Hha3 PLA2 (similar binding to other PLA2s but no neutralization). It seems that the only confirmed example of neutralization breadth shown is PLA2N (*M. fulvius*) and Nn19 (*N. nigricollis*); and of DH (*M. diastema*) since this venom was not part of the immunization. It would have been nice to see binding (BLI) and *in vitro* neutralization data for the 18 venoms used for immunization and comparison with other ‘broadly neutralizing’ mAbs in the literature.

Response: While the antibody library was indeed made with entirely different venoms, we do not think this would be relevant to do, since our phage display campaigns only focused on coral snake toxins and an evaluation of the library beyond that point was not attempted. Due to the complexity of each of the venoms used for immunization (containing between 20 and 80 different protein components), performing BLI and *in vitro* neutralization against each of them would be a very challenging and not necessarily informative task.

The term broadly-neutralizing here refers to the VHHs ability to both bind and neutralize toxins from different snake species and for different similar toxins within a single snake venom. We now see how this term could be confusing in some contexts, so we have substituted it for the term “cross-neutralizing” when referring to the discovered VHHs, to clarify that we expect them to neutralize more than one toxin.

Regarding the comparison with Coralmyx, I defer to more qualified reviewers as to whether this is notable or not in the context of this field. I was uncertain whether *M. fulvius* and *M. diastema* venoms provide a full and fair comparison, since the VHHs were established during the workup to neutralize PLA2/scNT respectively from these species; Coralmyx is produced using *M. nigrocinctus* venom, which was not tested, and I’m not sure what is known regarding its neutralization breadth. The points I made previously stand (that other combinations of 2 or more VHHs, and different molar ratios, could be superior). It is not so

clear to me that designing anti-venom mixtures based on component toxin abundance is automatically going to achieve the best result – and this hypothesis was not tested here.

Response:

Regarding the neutralization breadth of Coralmyn, there is limited information, but it has been shown to be effective against *M. fulvius* and *M. tener*, both North American coral snake species with a high abundance of PLA₂s (<https://doi.org/10.1016/j.toxicon.2007.10.004>). It has also been shown to be ineffective against venoms with a high 3FTx abundance, such as *M. surinamensis* (<https://doi.org/10.1081/CLT-120030943>), which is in accordance with the lack of neutralization observed for *M. diastema* in our work, and also highlights the advantage of oligoclonal recombinant antibody, or V_HH mixtures, as these can be designed to target also poorly immunogenic toxins, like 3FTxs.

Our work represents the first attempt on neutralization of coral snake venoms with defined oligoclonal V_HH mixtures and we therefore considered toxin abundance to maximize the chances of neutralization. We believe further optimization could potentially lead to the design of more optimal V_HH mixtures. However, since the venom composition varies between snake species, the optimal V_HH composition will likely vary between venoms and a final product would need to be a mixture that neutralizes multiple venoms. Therefore, optimizing the mixture on an individual venom does not make sense. Instead, here we show that a mixture can be used for neutralization, while the optimal final product composition is outside the scope of the project, but something a potential antivenom manufacturer should investigate before formulating the final recombinant antivenom product. We have now rephrased to clarify in lines 338-340.

Regarding the authors' statements that "this new work represents the first time that a monoclonal antibody mixture can neutralize the lethality of more complex venoms where the clinical syndrome is caused by more than one toxin class" and "neutralization of a complete venom with an oligoclonal mixture of only two mAb is expected to be viewed as both a surprising and highly relevant achievement in the field of toxinology," I defer to more qualified reviewers. This sounds like it could be a potentially useful contribution, but as a non-expert in toxinology I assumed this had already been demonstrated, given that these 2 toxins are responsible for much of the clinical syndrome. I did not conduct an exhaustive literature search but synergy between pAbs against PLA₂ and scNT (PMID 33423840) seems to be out there. If this is the major contribution of the paper it seems to me that this could be made more clear throughout, especially in the title and abstract.

Response:

Neutralization of complete venoms with polyclonal antivenoms has been achieved for coral snake venoms before and is the standard for antivenom development. The study cited by the reviewer is on equine polyclonal sera and is therefore not vis-à-vis comparable to our approach. Nonetheless, it shows that for many coral snake venoms, including *M. diastema*, it is necessary to obtain antibodies against short chain α NTxs to achieve complete neutralization, so we have added a reference to these findings in lines 267 to 270.

We have now also emphasized in the abstract that the main contribution of the paper is to achieve neutralization of lethality of whole venoms using monoclonal nanobodies instead of polyclonal sera from hyperimmunization.

Regarding minor point 7, it would be helpful to provide sequence alignments and % identity

for all PLA2 and scNT used in the study. Don't PLA2 from *M. fulvius* and *N. nigricollis* share about 70% identity?

Response:

We thank the reviewer for this suggestion. We have now complemented the sequence alignment in Supplementary Figure 1B.

Regarding minor point 8, do the authors have an explanation as to why VHH monomer (15 kDa) and Fc fusion (80 kDa) behave similarly but bivalent VHH (30 kDa) is much less effective in Fig. 5? This seems rather odd.

Response:

We agree with the reviewer that these results were unexpected and do not have a clear explanation for this. Since no improvement of survival was seen with any of the bivalent constructs, no further experiments were made with these. We have clarified this in line 294. When we examine the various constructs in BLI with toxin immobilized on the sensor followed by addition of the various V_{HH} constructs (Supplementary Figure 7), we do see an avidity effect of the bivalent constructs indicating that both binding sites of the molecules are available for antigen binding (lines 285-288).

In addition, we have evaluated the binding of the toxin DH to the monovalent and bivalent constructs immobilized on the BLI sensors, showing that the affinity is similar between the two constructs, as shown in the following:

A.) Monovalent TPL0629_01_D11

B.) Bivalent TPL0629_01_D11

V _{HH} construct	Toxin	KD (M)	KD Error	kon(1/Ms)	kon Error	kdis(1/s)	kdis Error	RMax	RMax Error	Full R ²
Monovalent	αNTx DH	2,37E-09	1,44E-11	2,08E+05	8,64E+02	4,94E-04	2,20E-06	0,3829	0,0003	0,9963
Bivalent	αNTx DH	3,10E-09	1,58E-11	2,18E+05	8,68E+02	6,77E-04	2,16E-06	0,2833	0,0002	0,9968

It is worth noting that an *in vivo* rescue experiment is complex and depends on both the toxins and the V_{HH}s pharmacokinetics, which likely is different between the bivalent V_{HH} compared to the V_{HH}-Fc, which we have now included in line 399-401. Neurotoxicity is very concentration-dependent and small differences in neutralizing capacity might have a huge influence on the survival of the mice (as the dose-response is a very steep curve, where the difference between life and death for mice can be very narrow). The bivalent constructs are tested at a similar molar ratio as the monovalent construct in terms of toxins to binding sites which corresponds to half toxin to whole molecules ratio. We speculate that the bivalent V_{HH} might result in just too few molecules in circulation after some time compared to the V_{HH}-Fc

which has a longer half-life due to the Fc region, and that the monovalent V_HHs starting with a higher number of molecules giving it an advantage.

Reviewer #2 (Remarks to the Author):

The authors have addressed all the points adequately.

Response:

We thank the reviewer for their time.

Reviewer #3 (Remarks to the Author):

The authors have sufficiently addressed my queries. My only suggestion now is to add statistics to their survival curve figures.

Response: We thank the reviewer for this suggestion and have implemented statistical analysis accordingly (Figures 4, 5, and 6; lines 738-743).

REVIEWERS' COMMENTS

Reviewer #1 (Remarks to the Author):

Since this manuscript has been sent to me for review once more I will do my best to deliver useful feedback. For full transparency I was and am OK with whatever decision the editor renders regarding incorporating any of this input.

Reviewer comment 1

“There is no argument that the authors have generated VHHs that bind and neutralize PLA2 and α NTx present in at least some venoms; however I believe neutralizing mAbs against these targets have been raised previously by other groups. I do not know how the VHHs compare with other mAbs or with pAb-based antivenoms in terms of binding/neutralization breadth.”

Author response: We agree with the reviewer that mAbs have been previously generated against similar venom components, although, to the best of our knowledge, never particularly against coral snake venom toxins. So far, we have not compared our VHHs with other VHHs, since there are no relevant VHHs to compare with. However, we have compared the discovered VHH mixtures with Coralmyx, the current treatment for Coral snake envenomation.

Reviewer appraisal: It seems to me that mAbs have been generated against coral snake PLA2 (Correa-Netta et al., 2023) and α NT (Tremeau et al., 1986). These are just two examples; I am not an expert in this literature but the authors are, there are also cobra toxin mAbs whose cross-reactivity to coral snake has not been evaluated. The VHHs described here have not been compared with coral snake toxin mAbs and their Fab fragments, which seems very relevant despite the minor distinction between a VHH and a Fab. While the VHH mixture is compared with Coralmyx, this is limited to in vivo neutralization of whole venom from the two species (*M. fulvius* and *M. diastema*) for which the VHHs were established to bind and neutralize the constituent toxins. A comparison across all coral snake species including the venom used to produce Coralmyx (*M. nigrocinctus*) might look quite different.

Recommended action: There is no need for any action or further discussion of this point as I understand the journal accepts uncertainty in the cross-reactivity of these antibodies vs. others as a limitation of the study.

Reviewer comment 2

“Regarding the multiplex immunization, while I understand that the goal was to provide wide coverage, it remains possible that immunization with one or a few venoms could have been as or more successful.”

Author response: We believe it is relevant to mention the rationale behind using multiplex immunization but agree that we cannot conclude that it was better than immunizing with individual venoms. We have rephrased this in line 85-88.

Reviewer appraisal: If the authors cannot conclude, based on their own or others' data, that multiplex immunization was better than immunizing with individual venoms, what is the rationale? The statement in line 85-88 is “that it has previously been shown that the use of

complex immunogen mixtures can be a good strategy for the generation of broadly-neutralizing polyclonal sera.” The paper cited (ref. 30) to support this statement does not really make this point – the thrust is that mixtures of partially purified toxins may be superior to mixtures of whole venoms at eliciting pAbs that neutralize venoms from the target species used in the immunization, since the antibody response is expected to be more focused on toxins rather than irrelevant proteins. Notably, for alpaca/llama immunization the authors used whole venoms; so based on ref. 30, maybe a greater diversity of VHHs could have been recovered if enriched toxin fractions had been used instead? In any case, it is self-evident that multiplex immunization is a good strategy to generate pAbs that recognize multiple distinct proteins (e.g., EGFR, SARS-CoV-2 spike and maltose binding protein) -- due to elicitation of individual mAbs that bind one of the targets – but this has no relevance to the generation of individual mAbs that cross-react with sequence-diverse proteins, which was the goal of this manuscript.

Suggested action: Delete the statement on line 85-88 and do not mention a rationale/strategy.

Reviewer comment 3

“Regarding the recombinant consensus toxin scNTx, failure to recover VHHs to two recombinant native toxins (rEury and rDH) is not very strong evidence that scNTx played an important role. Perhaps the better panning result with scNTx could be due to scNTx sharing greater homology than rDH in key regions with the scNTs within the venoms used for immunization, or at least those that were most immunogenic? There is no evidence that the scNTx was more useful than one or more recombinant native scNT(s) matched to the immunizing venoms – these experiments were not done.”

Author response: We agree with the reviewer that we have not thoroughly investigated if it is the consensus nature of scNTx that makes it a better phage display antigen in our experiment. We have rephrased to clarify this in lines 155-157. Nonetheless, our group has recently published a paper showing that the use of consensus toxins can be beneficial to discover more broadly neutralizing antibodies (<https://onlinelibrary.wiley.com/doi/full/10.1002/pro.4901>), so we have reformulated our text to provide the rationale, but done our best to avoid saying that this approach is the most optimal one to use.

Reviewer appraisal: The evidence in the paper cited that the consensus toxin approach works is very preliminary for several reasons. More importantly, this was a study of a naïve scFv library and whether any lessons learned are generalizable to antigen recognition by immune antibodies is highly uncertain.

Suggested action: Delete the statement on lines 91-92 and do not mention a rationale/strategy. The statement on lines 53-54 about the scNTx seems OK to me. For lines 155-157, revise to “compared to the two native coral snake toxins used in the experiment (rEury and rDH)” as other native toxins were not tested.

Reviewer comment 4

“Regarding cross-binding/breadth of neutralization... it seems that the only confirmed

example of neutralization breadth shown is PLA2N (*M. fulvius*) and Nn19 (*N. nigricollis*); and of DH (*M. diastema*) since this venom was not part of the immunization. It would have been nice to see binding (BLI) and in vitro neutralization data for the 18 venoms used for immunization. If it is not yet clear whether the neutralization breadth of the VHHs described here is notable or unexpected compared with other mAbs, I would argue that the term “broadly neutralizing” is not appropriate to use.”

Author response: While the antibody library was indeed made with entirely different venoms, we do not think this would be relevant to do, since our phage display campaigns only focused on coral snake toxins and an evaluation of the library beyond that point was not attempted. Due to the complexity of each of the venoms used for immunization (containing between 20 and 80 different protein components), performing BLI and in vitro neutralization against each of them would be a very challenging and not necessarily informative task.

The term broadly-neutralizing here refers to the VHHs ability to both bind and neutralize toxins from different snake species and for different similar toxins within a single snake venom. We now see how this term could be confusing in some contexts, so we have substituted it for the term “cross-neutralizing” when referring to the discovered VHHs, to clarify that we expect them to neutralize more than one toxin.

Reviewer appraisal: Yes, the immunization was with different venoms vs. the phage display selection targets, but the only documented mAb cross-reactivity is between Nn19 (*N. nigricollis*, possibly the immunizing venom against which the mAb was raised) and PLA2N (*M. fulvius*, selection antigen) and between an unknown aNT (immunizing venom) and DH (*M. diastema*, selection antigen). Patterns of binding and neutralization against the immunizing toxins/venoms as well as those of additional coral snakes would seem highly relevant as this would directly inform the clinical relevance of the VHH mixture. I did not understand how ‘broadly neutralizing’ could be used here to refer to the ability to neutralize different similar toxins within a single snake venom, since this is not applicable to coral snake venoms if I understand correctly, hopefully I have not missed something. The VHHs here were not shown to neutralize multiple different PLAs or aNTs within the venom from a single species.

Suggested action: There is no need for further experiments as I understand the journal accepts uncertainty in the cross-reactivity of these antibodies vs. others as a limitation of the study. Avoiding ‘broadly neutralizing’ seems like a good approach although there are a few cases where the language could still be improved. On line 177 change “broad cross-reactivity” to “cross-reactive binding” to avoid any confusion. On line 408, “are more broadly neutralizing” should be changed to “cross-neutralized venoms from *M. fulvius* and *M. diastema*, unlike the currently used antivenom Coralmyx which was only able to neutralize venom from *M. fulvius*” (see also Reviewer comment 5). On line 446 change “broadly neutralizing VHHs” to “cross-neutralizing VHHs”.

Reviewer comment 5

“Regarding the comparison with Coralmyx... I was uncertain whether *M. fulvius* and *M. diastema* venoms provide a full and fair comparison, since the VHHs were established during the workup to neutralize PLA2/scNT respectively from these species; Coralmyx is produced using *M. nigrocinctus* venom, which was not tested, and I’m not sure what is known regarding its neutralization breadth.”

Author response: Regarding the neutralization breadth of Coralmyn, there is limited information, but it has been shown to be effective against *M. fulvius* and *M. tener*, both North American coral snake species with a high abundance of PLA2s (<https://doi.org/10.1016/j.toxicon.2007.10.004>). It has also been shown to be ineffective against venoms with a high 3FTx abundance, such as *M. surinamensis* (<https://doi.org/10.1081/CLT-120030943>), which is in accordance with the lack of neutralization observed for *M. diastema* in our work, and also highlights the advantage of oligoclonal recombinant antibody, or VHH mixtures, as these can be designed to target also poorly immunogenic toxins, like 3FTxs.

Reviewer appraisal: It may or may not be true that Coralmyn is ineffective against venoms with a high 3FTx abundance – this statement is not made by the listed paper. Nevertheless, we expect that for both Coralmyn and the VHH mixture, breadth will depend on the extent to which binding affinity/neutralization potency remain sufficiently high for the PLA and 3FTx toxin constituents of venoms derived from diverse species. This has not been tested, but since Coralmyn was generated against a coral snake venom there is at least a possibility that it may neutralize coral snake PLAs more broadly than the PLA2 VHH, which was elicited against a non-coral snake PLA (possibly *N. nigricollis*). Even for *M. fulvius* venom (Fig. 6A), 100% protection for both Coralmyn and the VHH mixture does not preclude important potency differences that might become evident in more stringent models.

Suggested action: Remove statements suggesting superior breadth of the VHH mixture over Coralmyn on line 100, 336-337, 408, and anywhere else that I missed. On a more minor note, there is a footnote in Fig. 5 referring to statistical significance vs. Coralmyn which I think must be a mistake since it was not tested in this experiment.

Reviewer comment 6

“It is not so clear to me that designing anti-venom mixtures based on component toxin abundance is automatically going to achieve the best result.”

Author response: Our work represents the first attempt on neutralization of coral snake venoms with defined oligoclonal VHH mixtures and we therefore considered toxin abundance to maximize the chances of neutralization.

Reviewer appraisal: As far as I can tell there is no evidence that this approach maximized the chance of neutralization. It assumes that toxins are equally difficult to neutralize based on abundance and that antibodies within the mixture are equally potent at neutralizing different toxins, neither of which is likely to be true. While certainly a million to one molar ratio of VHHs would probably be a bad idea, I do not see why molar ratios of 1:1, 1:2, 1:3 or even further down this line of thinking could not have been as or more effective.

Suggested action: Avoid words like ‘design’ on lines 9, 59, 82, 131, 314, 335, 347, 446 which suggest that this approach was based on a rationale/strategy (something like ‘prepare’ or ‘select VHHs for’ the mixture would be more appropriate). It would be fine in the discussion to address the issue of mAb abundance ratios in oligoclonal cocktails for envenomation, although I suspect there is not much literature here. If you choose to do so, the assumptions underlying matching VHH molarity to toxin abundance should be mentioned and it should be clearly stated that the optimal ratio would need to be determined empirically

and is unlikely to depend solely on toxin abundance.

Reviewer comment 7

Regarding the authors' statements that "this new work represents the first time that a monoclonal antibody mixture can neutralize the lethality of more complex venoms where the clinical syndrome is caused by more than one toxin class" and "neutralization of a complete venom with an oligoclonal mixture of only two mAb is expected to be viewed as both a surprising and highly relevant achievement in the field of toxinology," I defer to more qualified reviewers. This sounds like it could be a potentially useful contribution, but as a non-expert in toxinology I assumed this had already been demonstrated, given that these 2 toxins are responsible for much of the clinical syndrome."

Author response: We have now also emphasized in the abstract that the main contribution of the paper is to achieve neutralization of lethality of whole venoms using monoclonal nanobodies instead of polyclonal sera from hyperimmunization.

[Also, from earlier in the rebuttal] In this work, we demonstrate that a venom with more than one toxic component can be neutralized by mixing antibodies and, also, show that this is feasible to do with a smaller antibody format, such as VHHs compared to earlier work where IgGs have been used (see e.g., Khalek et al. Science Translational Medicine 2024 or Ledsgaard et al. Nature Communications 2023). Neutralization of a complete snake venom often requires that the VHHs cross-neutralize multiple similar toxins, since a single venom normally has more than one toxin of each subfamily. We have clarified this in lines 312 to 314.

Reviewer appraisal: I appreciate now that the incremental advances here are: (1) that the VHH mixture targets 2 separate toxin types (e.g., vs. Lausten 2018 where a 3 mAb cocktail neutralized different dendrotoxins in black mamba venom but only by the i.c.v. route where alpha neurotoxins don't play much of a role), and (2) extension of the mAb cocktail approach to coral snakes. However, whether the mAb is a full IgG or fragment such as Fab or VHH is used seems relatively unimportant -- it is no special accomplishment to substitute VHHs as antivenoms instead of the Fab, Fab'2 or scFv fragments derived from IgGs, or IgGs themselves. VHH-Fcs are recognized as generally analogous to IgGs for most applications, and there is a significant body of literature on use of VHHs as antivenoms (since these do not require long half lives, and indeed most are Fab or Fab'2). The authors' own data (Fig. 5) suggest similar performance of VHH and VHH-Fc against aNT.

Suggested action: Revise lines 376-377 and statements elsewhere (did not check carefully) to clarify that the main advance here doesn't have to do with use of VHHs but is instead related to the demonstration that coral snake venom can be neutralized by 2 mAbs/VHHs, one against PLA and one against aNT.

Reviewer comment 8

"It would be helpful to provide sequence alignments and % identity for all PLA2 and scNT used in the study. Don't PLA2 from *M. fulvius* and *N. nigricollis* share about 70% identity?"

Author response: We thank the reviewer for this suggestion. We have now complemented the sequence alignment in Supplementary Figure 1B.

Reviewer appraisal: While this is an improvement, ideally the relevant toxins from venoms from all 18 species used for immunization plus the 5 toxins used in phage display selections plus the unrelated toxins in Fig. S5/S6 (*E. pyramidum*, *N. kaouthia*) would be shown along with their pairwise % identities. It appears that *N. nigricollis* PLA (immunizing) shares about 70% identity with *M. fulvius* PLA (selection) and that *N. haje* aNT (immunizing) share about 80% identity with the consensus toxin (selection) and with *M. diastema* aNT. This conflicts somewhat with the authors' earlier statement that "we are quite sure that it is not easy to achieve broad-neutralization of different toxins from the same family, as the sequence similarity may be below 40%." It remains unclear whether other immunizing PLAs or aNTs may share higher identity with *M. fulvius* PLA and *M. diastema* aNT than this. Interestingly *N. nigricollis* PLA also shares about 70% identity with *H. haemachatus* PLA which was enough to achieve binding but not neutralization.

Suggested action: Consider completing the alignments and adding % identity. At the minimum the sequences of *M. fulvius* aNT and *M. diastema* PLA should be added, since presumably the relevant VHHs must cross-neutralize these, right? Do the authors assume that VHH TPL0629_01_D11 binds and neutralizes the former while VHH TPL0637_01_A07 binds and neutralizes the latter, despite this not having been tested? This should be explicitly stated. Also consider commenting somewhere (perhaps in the discussion) on the level of conservation between immunizing/selection antigens as well as between toxins for which cross-reactivity and cross-neutralization occur and do not occur (e.g. for *H. hemachatus* PLA); what are the implications for ability to neutralize venoms from different coral snakes, or would this be on a case-by-case basis?

Reviewer comment 9

"Do the authors have an explanation as to why VHH monomer (15 kDa) and Fc fusion (80 kDa) behave similarly but bivalent VHH (30 kDa) is much less effective in Fig. 5?"

Author response: We agree with the reviewer that these results were unexpected and do not have a clear explanation for this. It is worth noting that an in vivo rescue experiment is complex and depends on both the toxins and the VHHs pharmacokinetics, which likely is different between the bivalent VHH compared to the VHH-Fc.

Reviewer appraisal: Reasonable to state this finding is unexpected and could have multiple explanations after binding problems were ruled out as a precaution.

Suggested action: None.

REVIEWERS' COMMENTS

Reviewer #1 (Remarks to the Author):

Since this manuscript has been sent to me for review once more I will do my best to deliver useful feedback. For full transparency I was and am OK with whatever decision the editor renders regarding incorporating any of this input.

Answer: We thank the reviewer for their thorough revision of our work. Please find our responses to each of the remaining comments below.

Reviewer comment 1

“There is no argument that the authors have generated VHHs that bind and neutralize PLA2 and α NTx present in at least some venoms; however I believe neutralizing mAbs against these targets have been raised previously by other groups. I do not know how the VHHs compare with other mAbs or with pAb-based antivenoms in terms of binding/neutralization breadth.”

Author response: We agree with the reviewer that mAbs have been previously generated against similar venom components, although, to the best of our knowledge, never particularly against coral snake venom toxins. So far, we have not compared our VHHs with other VHHs, since there are no relevant VHHs to compare with. However, we have compared the discovered VHH mixtures with Coralmyx, the current treatment for Coral snake envenomation.

Reviewer appraisal: It seems to me that mAbs have been generated against coral snake PLA2 (Correa-Netta et al., 2023) and α NT (Tremeau et al., 1986). These are just two examples; I am not an expert in this literature but the authors are, there are also cobra toxin mAbs whose cross-reactivity to coral snake has not been evaluated. The VHHs described here have not been compared with coral snake toxin mAbs and their Fab fragments, which seems very relevant despite the minor distinction between a VHH and a Fab. While the VHH mixture is compared with Coralmyx, this is limited to in vivo neutralization of whole venom from the two species (*M. fulvius* and *M. diastema*) for which the VHHs were established to bind and neutralize the constituent toxins. A comparison across all coral snake species including the venom used to produce Coralmyx (*M. nigrocinctus*) might look quite different.

Recommended action: There is no need for any action or further discussion of this point as I understand the journal accepts uncertainty in the cross-reactivity of these antibodies vs. others as a limitation of the study.

Answer: No further action taken.

Reviewer comment 2

“Regarding the multiplex immunization, while I understand that the goal was to provide wide coverage, it remains possible that immunization with one or a few venoms could

have been as or more successful.”

Author response: We believe it is relevant to mention the rationale behind using multiplex immunization but agree that we cannot conclude that it was better than immunizing with individual venoms. We have rephrased this in line 85-88.

Reviewer appraisal: If the authors cannot conclude, based on their own or others' data, that multiplex immunization was better than immunizing with individual venoms, what is the rationale? The statement in line 85-88 is “that it has previously been shown that the use of complex immunogen mixtures can be a good strategy for the generation of broadly-neutralizing polyclonal sera.” The paper cited (ref. 30) to support this statement does not really make this point – the thrust is that mixtures of partially purified toxins may be superior to mixtures of whole venoms at eliciting pAbs that neutralize venoms from the target species used in the immunization, since the antibody response is expected to be more focused on toxins rather than irrelevant proteins. Notably, for alpaca/llama immunization the authors used whole venoms; so based on ref. 30, maybe a greater diversity of VHHs could have been recovered if enriched toxin fractions had been used instead? In any case, it is self-evident that multiplex immunization is a good strategy to generate pAbs that recognize multiple distinct proteins (e.g., EGFR, SARS-CoV-2 spike and maltose binding protein) -- due to elicitation of individual mAbs that bind one of the targets – but this has no relevance to the generation of individual mAbs that cross-react with sequence-diverse proteins, which was the goal of this manuscript.

Suggested action: Delete the statement on line 85-88 and do not mention a rationale/strategy.

Answer: The statement has been deleted.

Reviewer comment 3

“Regarding the recombinant consensus toxin scNTx, failure to recover VHHs to two recombinant native toxins (rEury and rDH) is not very strong evidence that scNTx played an important role. Perhaps the better panning result with scNTx could be due to scNTx sharing greater homology than rDH in key regions with the scNTs within the venoms used for immunization, or at least those that were most immunogenic? There is no evidence that the scNTx was more useful than one or more recombinant native scNT(s) matched to the immunizing venoms – these experiments were not done.”

Author response: We agree with the reviewer that we have not thoroughly investigated if it is the consensus nature of scNTx that makes it a better phage display antigen in our experiment. We have rephrased to clarify this in lines 155-157. Nonetheless, our group has recently published a paper showing that the use of consensus toxins can be beneficial to discover more broadly neutralizing antibodies (<https://onlinelibrary.wiley.com/doi/full/10.1002/pro.4901>), so we have reformulated our text to provide the rationale, but done our best to avoid saying that this approach is

the most optimal one to use.

Reviewer appraisal: The evidence in the paper cited that the consensus toxin approach works is very preliminary for several reasons. More importantly, this was a study of a naïve scFv library and whether any lessons learned are generalizable to antigen recognition by immune antibodies is highly uncertain.

Suggested action: Delete the statement on lines 91-92 and do not mention a rationale/strategy. The statement on lines 53-54 about the scNTx seems OK to me. For lines 155-157, revise to “compared to the two native coral snake toxins used in the experiment (rEury and rDH)” as other native toxins were not tested.

Answer: The statement in lines 91-92 has been deleted and the one in line 167 has been rephrased

Reviewer comment 4

“Regarding cross-binding/breadth of neutralization... it seems that the only confirmed example of neutralization breadth shown is PLA2N (*M. fulvius*) and Nn19 (*N. nigricollis*); and of DH (*M. diastema*) since this venom was not part of the immunization. It would have been nice to see binding (BLI) and in vitro neutralization data for the 18 venoms used for immunization. If it is not yet clear whether the neutralization breadth of the VHHs described here is notable or unexpected compared with other mAbs, I would argue that the term “broadly neutralizing” is not appropriate to use.”

Author response: While the antibody library was indeed made with entirely different venoms, we do not think this would be relevant to do, since our phage display campaigns only focused on coral snake toxins and an evaluation of the library beyond that point was not attempted. Due to the complexity of each of the venoms used for immunization (containing between 20 and 80 different protein components), performing BLI and in vitro neutralization against each of them would be a very challenging and not necessarily informative task.

The term broadly-neutralizing here refers to the VHHs ability to both bind and neutralize toxins from different snake species and for different similar toxins within a single snake venom. We now see how this term could be confusing in some contexts, so we have substituted it for the term “cross-neutralizing” when referring to the discovered VHHs, to clarify that we expect them to neutralize more than one toxin.

Reviewer appraisal: Yes, the immunization was with different venoms vs. the phage display selection targets, but the only documented mAb cross-reactivity is between Nn19 (*N. nigricollis*, possibly the immunizing venom against which the mAb was raised) and PLA2N (*M. fulvius*, selection antigen) and between an unknown aNT (immunizing venom) and DH (*M. diastema*, selection antigen). Patterns of binding and neutralization against the immunizing toxins/venoms as well as those of additional coral snakes would seem highly relevant as this would directly inform the clinical relevance of the

VHH mixture. I did not understand how ‘broadly neutralizing’ could be used here to refer to the ability to neutralize different similar toxins within a single snake venom, since this is not applicable to coral snake venoms if I understand correctly, hopefully I have not missed something. The VHHs here were not shown to neutralize multiple different PLAs or aNTs within the venom from a single species.

Suggested action: There is no need for further experiments as I understand the journal accepts uncertainty in the cross-reactivity of these antibodies vs. others as a limitation of the study. Avoiding ‘broadly neutralizing’ seems like a good approach although there are a few cases where the language could still be improved. On line 177 change “broad cross-reactivity” to “cross-reactive binding” to avoid any confusion. On line 408, “are more broadly neutralizing” should be changed to “cross-neutralized venoms from *M. fulvius* and *M. diastema*, unlike the currently used antivenom Coralmyx which was only able to neutralize venom from *M. fulvius*” (see also Reviewer comment 5). On line 446 change “broadly neutralizing VHHs” to “cross-neutralizing VHHs”.

Answer: The terms in lines 187 and 462 have been changed and the statement in line 408 has been modified according to the reviewer’s suggestions.

Reviewer comment 5

“Regarding the comparison with Coralmyx... I was uncertain whether *M. fulvius* and *M. diastema* venoms provide a full and fair comparison, since the VHHs were established during the workup to neutralize PLA2/scNT respectively from these species; Coralmyx is produced using *M. nigrocinctus* venom, which was not tested, and I’m not sure what is known regarding its neutralization breadth.”

Author response: Regarding the neutralization breadth of Coralmyx, there is limited information, but it has been shown to be effective against *M. fulvius* and *M. tener*, both North American coral snake species with a high abundance of PLA2s (<https://doi.org/10.1016/j.toxicon.2007.10.004>). It has also been shown to be ineffective against venoms with a high 3FTx abundance, such as *M. surinamensis* (<https://doi.org/10.1081/CLT-120030943>), which is in accordance with the lack of neutralization observed for *M. diastema* in our work, and also highlights the advantage of oligoclonal recombinant antibody, or VHH mixtures, as these can be designed to target also poorly immunogenic toxins, like 3FTxs.

Reviewer appraisal: It may or may not be true that Coralmyx is ineffective against venoms with a high 3FTx abundance – this statement is not made by the listed paper. Nevertheless, we expect that for both Coralmyx and the VHH mixture, breadth will depend on the extent to which binding affinity/neutralization potency remain sufficiently high for the PLA and 3FTx toxin constituents of venoms derived from diverse species. This has not been tested, but since Coralmyx was generated against a coral snake venom there is at least a possibility that it may neutralize coral snake PLAs more broadly than the PLA2 VHH, which was elicited against a non-coral snake PLA (possibly *N. nigricollis*). Even for *M. fulvius* venom (Fig. 6A), 100% protection for both Coralmyx

and the VHH mixture does not preclude important potency differences that might become evident in more stringent models.

Suggested action: Remove statements suggesting superior breadth of the VHH mixture over Coralmyx on line 100, 336-337, 408, and anywhere else that I missed. On a more minor note, there is a footnote in Fig. 5 referring to statistical significance vs. Coralmyx which I think must be a mistake since it was not tested in this experiment.

Answer: Statements in lines 108, 352, and 425 have been modified accordingly. The footnote in Fig. 5 was indeed a mistake and has now been removed, we thank the reviewer for noticing it.

Reviewer comment 6

“It is not so clear to me that designing anti-venom mixtures based on component toxin abundance is automatically going to achieve the best result.”

Author response: Our work represents the first attempt on neutralization of coral snake venoms with defined oligoclonal VHH mixtures and we therefore considered toxin abundance to maximize the chances of neutralization.

Reviewer appraisal: As far as I can tell there is no evidence that this approach maximized the chance of neutralization. It assumes that toxins are equally difficult to neutralize based on abundance and that antibodies within the mixture are equally potent at neutralizing different toxins, neither of which is likely to be true. While certainly a million to one molar ratio of VHHs would probably be a bad idea, I do not see why molar ratios of 1:1, 1:2, 1:3 or even further down this line of thinking could not have been as or more effective.

Suggested action: Avoid words like ‘design’ on lines 9, 59, 82, 131, 314, 335, 347, 446 which suggest that this approach was based on a rationale/strategy (something like ‘prepare’ or ‘select VHHs for’ the mixture would be more appropriate). It would be fine in the discussion to address the issue of mAb abundance ratios in oligoclonal cocktails for envenomation, although I suspect there is not much literature here. If you choose to do so, the assumptions underlying matching VHH molarity to toxin abundance should be mentioned and it should be clearly stated that the optimal ratio would need to be determined empirically and is unlikely to depend solely on toxin abundance.

Answer: The word design has been replaced by the term prepare, or similar ones throughout the text.

Reviewer comment 7

Regarding the authors’ statements that “this new work represents the first time that a monoclonal antibody mixture can neutralize the lethality of more complex venoms where the clinical syndrome is caused by more than one toxin class” and

“neutralization of a complete venom with an oligoclonal mixture of only two mAb is expected to be viewed as both a surprising and highly relevant achievement in the field of toxinology,” I defer to more qualified reviewers. This sounds like it could be a potentially useful contribution, but as a non-expert in toxinology I assumed this had already been demonstrated, given that these 2 toxins are responsible for much of the clinical syndrome.”

Author response: We have now also emphasized in the abstract that the main contribution of the paper is to achieve neutralization of lethality of whole venoms using monoclonal nanobodies instead of polyclonal sera from hyperimmunization.

[Also, from earlier in the rebuttal] In this work, we demonstrate that a venom with more than one toxic component can be neutralized by mixing antibodies and, also, show that this is feasible to do with a smaller antibody format, such as VHHs compared to earlier work where IgGs have been used (see e.g., Khalek et al. Science Translational Medicine 2024 or Ledsgaard et al. Nature Communications 2023). Neutralization of a complete snake venom often requires that the VHHs cross-neutralize multiple similar toxins, since a single venom normally has more than one toxin of each subfamily. We have clarified this in lines 312 to 314.

Reviewer appraisal: I appreciate now that the incremental advances here are: (1) that the VHH mixture targets 2 separate toxin types (e.g., vs. Lausten 2018 where a 3 mAb cocktail neutralized different dendrotoxins in black mamba venom but only by the i.c.v. route where alpha neurotoxins don't play much of a role), and (2) extension of the mAb cocktail approach to coral snakes. However, whether the mAb is a full IgG or fragment such as Fab or VHH is used seems relatively unimportant -- it is no special accomplishment to substitute VHHs as antivenoms instead of the Fab, Fab'2 or scFv fragments derived from IgGs, or IgGs themselves. VHH-Fcs are recognized as generally analogous to IgGs for most applications, and there is a significant body of literature on use of VHHs as antivenoms (since these do not require long half lives, and indeed most are Fab or Fab'2). The authors' own data (Fig. 5) suggest similar performance of VHH and VHH-Fc against aNT.

Suggested action: Revise lines 376-377 and statements elsewhere (did not check carefully) to clarify that the main advance here doesn't have to do with use of VHHs but is instead related to the demonstration that coral snake venom can be neutralized by 2 mAbs/VHHs, one against PLA and one against aNT.

Answer: We respectfully disagree with the reviewer regarding the fact that the use of V_HHs for antivenoms is no special accomplishment. In the context of snakebite envenomation, the use of a thermostable molecule such as a V_HH instead of IgG-derived fragments would allow the distribution of the antivenom to regions where a cold chain cannot be maintained. This would be a major improvement from current antivenoms and therefore we believe the use of V_HHs in this work is a proof of concept definitely worth mentioning.

Reviewer comment 8

“It would be helpful to provide sequence alignments and % identity for all PLA₂ and scNT used in the study. Don't PLA₂ from *M. fulvius* and *N. nigricollis* share about 70% identity?”

Author response: We thank the reviewer for this suggestion. We have now complemented the sequence alignment in Supplementary Figure 1B.

Reviewer appraisal: While this is an improvement, ideally the relevant toxins from venoms from all 18 species used for immunization plus the 5 toxins used in phage display selections plus the unrelated toxins in Fig. S5/S6 (*E. pyramidum*, *N. kaouthia*) would be shown along with their pairwise % identities. It appears that *N. nigricollis* PLA (immunizing) shares about 70% identity with *M. fulvius* PLA (selection) and that *N. haje* aNT (immunizing) share about 80% identity with the consensus toxin (selection) and with *M. diastema* aNT. This conflicts somewhat with the authors' earlier statement that “we are quite sure that it is not easy to achieve broad-neutralization of different toxins from the same family, as the sequence similarity may be below 40%.” It remains unclear whether other immunizing PLAs or aNTs may share higher identity with *M. fulvius* PLA and *M. diastema* aNT than this. Interestingly *N. nigricollis* PLA also shares about 70% identity with *H. haemachatus* PLA which was enough to achieve binding but not neutralization.

Suggested action: Consider completing the alignments and adding % identity. At the minimum the sequences of *M. fulvius* aNT and *M. diastema* PLA should be added, since presumably the relevant VHHs must cross-neutralize these, right? Do the authors assume that VHH TPL0629_01_D11 binds and neutralizes the former while VHH TPL0637_01_A07 binds and neutralizes the latter, despite this not having been tested? This should be explicitly stated. Also consider commenting somewhere (perhaps in the discussion) on the level of conservation between immunizing/selection antigens as well as between toxins for which cross-reactivity and cross-neutralization occur and do not occur (e.g. for *H. hemachatus* PLA); what are the implications for ability to neutralize venoms from different coral snakes, or would this be on a case-by-case basis?

Answer: Unfortunately, complete sequences of relevant toxins from the 18 elapids used for immunization are not available, we have added available PLA₂ sequences with highest sequence similarity to PLA₂N, as well as others from various elapid species. This is not intended as a comprehensive list of PLA₂ sequences, which has now been stated in the legend of Supplementary Figure 1. Complete sequences are also not yet available in the case of *M. diastema* PLA₂s, since only the 3FTx fraction of the venom has been previously described. We have added the % identity values to Supplementary Figure 1.

We did not specifically test binding of TPL0629_01_D11 to *M. diastema* PLA₂s and TPL0637_01_A07 to *M. fulvius* 3FTxs. In our opinion, the fact that both V_HHs are

necessary for neutralization of whole venoms in the *in vivo* model, is strong evidence proving that both V_HHs bind and neutralize relevant lethal toxins in the two venoms. The identity of these toxins is not currently known.

Reviewer comment 9

“Do the authors have an explanation as to why VHH monomer (15 kDa) and Fc fusion (80 kDa) behave similarly but bivalent VHH (30 kDa) is much less effective in Fig. 5?”

Author response: We agree with the reviewer that these results were unexpected and do not have a clear explanation for this. It is worth noting that an *in vivo* rescue experiment is complex and depends on both the toxins and the VHHs pharmacokinetics, which likely is different between the bivalent VHH compared to the VHH-Fc.

Reviewer appraisal: Reasonable to state this finding is unexpected and could have multiple explanations after binding problems were ruled out as a precaution.

Suggested action: None.

Answer: No further action taken.